# TESSER: Transfer-Enhancing Adversarial Attacks from Vision Transformers via Spectral and Semantic Regularization

## Abstract

Adversarial transferability remains a critical challenge in evaluating the robustness of deep neural networks. In security-critical applications, transferability enables black-box attacks without access to model internals, making it a key concern for real-world adversarial threat assessment. While Vision Transformers (ViTs) have demonstrated strong adversarial performance, existing attacks often fail to transfer effectively across architectures, especially from ViTs to Convolutional Neural Networks (CNNs) or hybrid models. In this paper, we introduce **TESSER**, a novel adversarial attack framework that enhances transferability via two key strategies: (1) *Feature-Sensitive Gradient Scaling (FSGS)*, which modulates gradients based on token-wise importance derived from intermediate feature activations, and (2) *Spectral Smoothness Regularization (SSR)*, which suppresses high-frequency noise in perturbations using a differentiable Gaussian prior. These components work in tandem to generate perturbations that are both semantically meaningful and spectrally smooth. Extensive experiments on ImageNet across 14 diverse architectures demonstrate that TESSER achieves +10.9% higher attack succes rate (ASR) on CNNs and +7.2% on ViTs compared to the state-of-the-art Adaptive Token Tuning (ATT) method. Moreover, TESSER significantly improves robustness against defended models, achieving 53.55% ASR on adversarially trained CNNs and +15% higher ASR on robust ViTs. Qualitative analysis shows strong alignment between TESSER's perturbations and salient visual regions identified via Grad-CAM, while frequency-domain analysis reveals a 12% reduction in high-frequency energy, confirming the effectiveness of spectral regularization.

## 1 Introduction

Deep learning models, particularly Convolutional Neural Networks (CNNs) and Vision Transformers (ViTs), have achieved state-of-the-art performance across a broad spectrum of computer vision tasks (Carion et al., 2020; Zhu et al., 2021; Ma et al., 2022). Despite this progress, these models remain highly vulnerable to adversarial examples–carefully crafted perturbations that are imperceptible to humans but cause misclassification (Goodfellow et al., 2014; Guesmi et al., 2023; 2024a;b). In safety-critical applications such as autonomous driving and medical imaging, this fragility raises significant security concerns.

Although white-box attacks, where attackers have full access to model parameters, have been extensively studied, black-box settings are more realistic in practice. These are based on the principle of *transferability*, where adversarial examples generated on a surrogate model are expected to fool unseen target models. However, transferability across architectures, especially from ViTs to CNNs or hybrid models, remains limited due to two key challenges: (1) the lack of **semantic selectivity**, where all tokens are perturbed uniformly without considering their relevance to the model's prediction, and (2) the presence of **high-frequency noise** in perturbations, which tends to encode brittle, model-specific artifacts that do not generalize well.

Several recent works, such as ATT (Ming et al., 2024) and TGR (Zhang et al., 2023), have explored ViT-specific mechanisms for improving transferability by truncating or regularizing gradient flows. However, these approaches either use fixed gradient masks or overlook token-level semantics, leading

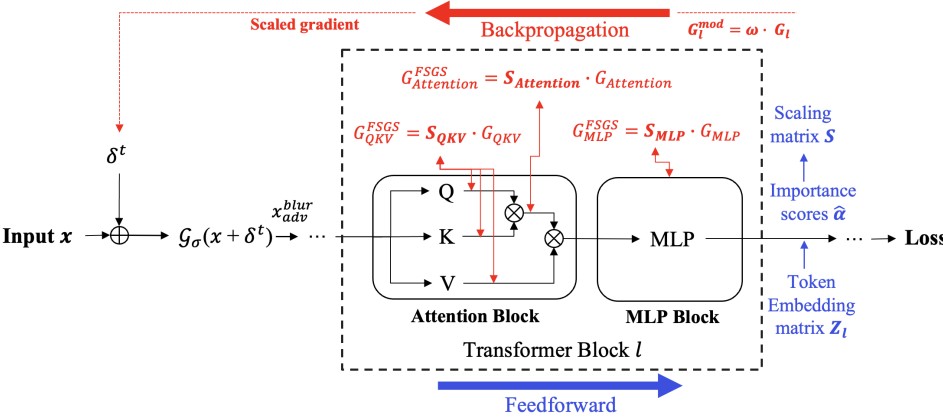

Figure 1: **Overview of the TESSER attack framework.** At each iteration, an adversarial perturbation $\delta^t$ is applied to the input image and smoothed via differentiable Gaussian blur $\mathcal{G}_\sigma(\cdot)$ to enforce spectral smoothness (SSR). The perturbed input is passed through the transformer, where token embeddings $Z_l$ from each layer are used to compute token-wise importance scores $\hat{\alpha}$, which in turn define gradient scaling masks $S$. During backpropagation, gradients for the Attention, QKV, and MLP modules are reweighted according to their respective scaling masks ($S_{\text{Attention}}, S_{\text{QKV}}, S_{\text{MLP}}$) using Feature-Sensitive Gradient Scaling (FSGS). This encourages perturbations to align with semantically meaningful and transferable features while suppressing noise and irrelevant gradients.

to suboptimal alignment with transferable visual features. In this paper, we introduce **TESSER** (*Transfer-Enhancing Semantic and Spectral Regularization*) a novel adversarial attack framework specifically designed to improve black-box transferability from ViT-based models to a diverse set of architectures. TESSER integrates two complementary strategies:

- *Feature-Sensitive Gradient Scaling (FSGS)*: a token-level gradient modulation method that scales gradients based on token importance derived from intermediate embeddings. Inspired by recent findings correlating token activation magnitudes with semantic relevance (Kobayashi et al., 2020; Wu et al., 2024; Modarressi et al., 2022), FSGS steers the attack toward semantically meaningful regions and away from background or non-informative tokens, enhancing cross-model generalization.

- *Spectral Smoothness Regularization (SSR)*: a lightweight regularization mechanism that applies a differentiable Gaussian blur during each optimization step. SSR suppresses high-frequency noise, promoting low-frequency perturbations that are more resilient across architectures, particularly beneficial when transferring to CNNs and adversarially trained models.

Together, these modules enable TESSER to produce perturbations that are semantically aligned and spectrally smooth, two characteristics that we empirically demonstrate to be critical for enhancing transferability in adversarial attacks. Our main contributions are summarized as follows:

- We propose **TESSER**, a novel adversarial attack framework that combines semantic- and spectral-aware regularization to improve transferability from ViTs.

- We introduce Feature-Sensitive Gradient Scaling (FSGS), which reweights gradients for Attention, QKV, and MLP modules based on token-level importance, encouraging semantically aligned perturbations.

- We incorporate Spectral Smoothness Regularization (SSR) to reduce high-frequency noise and enhance cross-architecture generalization.

- We conduct extensive experiments on ImageNet across 14 diverse models (including ViTs, CNNs, and adversarially defended ViTs and CNNs), demonstrating that TESSER achieves up to **+10.9%** higher ASR over state-of-the-art baselines and consistently outperforms existing attacks in both black-box and robust scenarios.

- We conduct comprehensive ablation studies, Grad-CAM-based semantic alignment evaluations (Section 4.4), and frequency-domain analyses (Section 4.5) to demonstrate both the effectiveness and interpretability of our approach.

## 2 RELATED WORK

**Adversarial Attacks on CNNs and ViTs.** Adversarial attacks are small, human-imperceptible perturbations intentionally added to input data to mislead deep learning models (Goodfellow et al., 2014). For Convolutional Neural Networks (CNNs), numerous gradient-based attacks have been proposed to improve transferability, including momentum-based methods (Dong et al., 2018a), variance tuning (Huang et al., 2019), and gradient skipping techniques (Wu et al., 2020). These methods aim to stabilize perturbation updates and avoid local optima in the input space. However, attack techniques designed for CNNs do not transfer well to Vision Transformers (ViTs), which have fundamentally different architectures and information flow patterns. Recent works have proposed ViT-specific attacks that exploit token structure and attention mechanisms (Naseer et al., 2022; Wei et al., 2022). For example, Token Gradient Regularization (TGR) (Zhang et al., 2023) modifies intermediate-layer gradients to reduce token-wise variance, improving transferability within ViT families.

Regularizing gradients is an effective way to suppress model-specific patterns and improve cross-model generalization. In CNNs, methods such as SGM (Wu et al., 2020) and BPA (Xiaosen et al., 2023) aim to manipulate the gradient flow through skip connections or rectify distortions introduced by nonlinearities. Others have employed gradient variance reduction (Huang et al., 2019) and ensemble-based tuning (Xiong et al., 2022). Attacks based on feature information (Wang et al., 2021; Ganeshan et al., 2019) focus on disrupting salient internal representations. However, improperly guided feature-based attacks risk discarding useful information and reducing transferability. To mitigate this, neuron attribution methods (Zhang et al., 2023) and attention map diversification (Ren et al., 2025) have been explored, particularly in ViTs. DiffAttack Chen et al. (2025) leverages generative diffusion models to craft adversarial examples, exploiting their ability to model natural image distributions. By iteratively guiding the diffusion process with adversarial objectives, it produces perturbations that are both transferable and perceptually realistic. Compared to gradient-based methods, DiffAttack introduces higher computational cost but demonstrates stronger performance in black-box and cross-architecture scenarios.

Forward Propagation Refinement (FPR) (Ren et al., 2025) is a recent surrogate-refinement strategy tailored for improving adversarial transferability on Vision Transformers (ViTs). Unlike prior methods that modify only the backward pass (e.g., PNAPO, TGR, GNS), FPR explicitly refines forward activations by diversifying attention maps and stabilizing token embeddings. Specifically, Attention Map Diversification (AMD) introduces controlled stochasticity into selected attention heads to mitigate surrogate overfitting and implicitly induce beneficial gradient vanishing, while Momentum Token Embedding (MTE) accumulates historical token embeddings to avoid local-optimum instability during attack iterations.

ATT (Ming et al., 2024) introduces hybrid token gradient truncation by weakening gradients in attention and QKV blocks across layers of a ViT model. It leverages empirical observations of gradient variance to suppress high-magnitude gradients associated with overfitting, thereby improving transferability. However, ATT applies static truncation and does not explicitly consider token-level semantic relevance, which may limit its effectiveness when generalizing across diverse architectures. In contrast, our method introduces *Feature-Sensitive Gradient Scaling (FSGS)*, which adaptively reweights gradients at a token level based on feature norms. This allows us to preserve semantically important gradients while suppressing noisy or architecture-specific ones, achieving improved transferability across ViTs, hybrids, and CNNs.

**Input Diversity and Spectral Regularization.** Input diversity has been widely adopted to improve adversarial transferability. DI-FGSM (Xie et al., 2019) applies random resizing and padding, while PatchOut (Wei et al., 2022) discards patch-wise perturbations to prevent overfitting. Recent self-paced extensions further refine patch discarding based on semantic guidance (Ming et al., 2024). While these approaches diversify the spatial patterns of inputs, few works address the frequency structure of perturbations. Our method incorporates *Spectral Smoothness Regularization (SSR)* by applying differentiable Gaussian blur during optimization. SSR suppresses high-frequency noise and promotes smooth perturbation patterns that generalize better across model architectures, particularly important for CNNs and early ViT layers that rely on localized features. **Importantly, input diversity is orthogonal to our method**, and can be combined with TESSER for further gains. We provide additional results and analysis combining input diversity with our framework in the Appendix C.

# 3 METHODOLOGY

## 3.1 PRELIMINARIES

Let $\mathbf{x} \in \mathbb{R}^{C \times H \times W}$ denote an input image with ground-truth label $y \in \{1, 2, \ldots, K\}$, and let $f(\cdot)$ be a deep neural network classifier. The goal of an untargeted adversarial attack is to generate a perturbation $\delta$ such that the perturbed input $\mathbf{x}^{\text{adv}} = \mathbf{x} + \delta$ is misclassified by the model, i.e., $f(\mathbf{x}^{\text{adv}}) \neq y$, while ensuring that $\|\delta\|_\infty \leq \epsilon$. Unlike CNNs that process local image regions hierarchically, Vision Transformers (Dosovitskiy et al., 2021) operate on a sequence of non-overlapping image patches. Given an input image $\mathbf{x}$, it is partitioned into $N = \frac{HW}{P^2}$ patches of size $P \times P$, each linearly projected to a $D$-dimensional embedding, resulting in tokens $\{\mathbf{z}_1, \ldots, \mathbf{z}_N\} \subset \mathbb{R}^D$. A learnable classification token $\mathbf{z}_{\text{cls}}$ is prepended, yielding a token sequence $\mathbf{Z}^{(0)} \in \mathbb{R}^{(N+1) \times D}$, which is enriched with positional encodings. ViTs consist of a stack of $L$ transformer blocks. Each block contains a Multi-Head Self-Attention (MHSA) module and a Multi-Layer Perceptron (MLP) module, connected via residual connections and layer normalization (LN).

## 3.2 FEATURE-SENSITIVE GRADIENT SCALING (FSGS)

To improve transferability, we propose *Feature-Sensitive Gradient Scaling (FSGS)*, a fine-grained gradient modulation strategy that steers adversarial updates toward semantically relevant tokens while suppressing gradients associated with model-specific or noisy patterns. Unlike prior methods such as ATT (Ming et al., 2024) and TGR (Zhang et al., 2023), which rely on fixed truncation or uniform regularization, FSGS leverages intermediate transformer features to dynamically adjust gradient flow on a per-token basis.

**Limitations of Prior Gradient Modulation Approaches.** ATT weakens gradients across transformer modules based on empirical variance, but applies static masks that may disregard salient tokens. TGR promotes token-wise gradient uniformity without regard for token semantics, leading to potentially ineffective or redundant updates. In contrast, FSGS introduces adaptive scaling conditioned on the importance of each token, measured directly from the model's internal activations. This content-aware reweighting enhances the alignment of perturbations with generalizable visual features and improves cross-architecture transfer.

**Why Token Activation Norm and Feature-Sensitive Gradient Scaling (FSGS)?** Token activation norms in Vision Transformers have been empirically shown to correlate with semantic saliency, with higher-norm tokens often corresponding to class-relevant features or foreground objects Kobayashi et al. (2020); Modarressi et al. (2022); Wu et al. (2024). Our Grad-CAM visualizations (Section 4.4) confirm this trend, showing strong alignment between high-norm tokens and semantically meaningful regions. This motivates using token norm as a saliency prior to guide adversarial perturbations. FSGS operationalizes this intuition by amplifying gradients from semantically important tokens while suppressing less informative ones. Importantly, not all layers benefit equally: early ViT layers capture low-level, architecture-dependent patterns (textures, positional cues) that hinder transfer, whereas deeper layers encode more robust, class-discriminative features Raghu et al. (2021); Bhojanapalli et al. (2021); Kim et al. (2024). To account for this, FSGS adopts a dual-stage strategy: in early layers, gradients are scaled by $(1 - \alpha)$ to downweight noisy signals, while in deeper layers, $\alpha$ is used to strengthen semantically aligned features. This design ensures perturbations are both semantically grounded and transferable across architectures, improving attack effectiveness in black-box settings (see Appendix A).

**Token-Level Importance Estimation.** Given a token embedding matrix $\mathbf{Z} \in \mathbb{R}^{T \times D}$, we estimate the importance of token $i$ using the activation norm $\alpha_i = \|\mathbf{z}_i\|_2$, which serves as a proxy for semantic saliency. This assumption is supported by prior work in both NLP and vision (Kobayashi et al., 2020; Wu et al., 2024; Modarressi et al., 2022), which shows that activation magnitudes often correlate with token informativeness or attention saliency. For instance, Kobayashi et al. (2020) and Modarressi et al. (2022) argue that vector norms contribute substantially to a token's influence, while Wu et al. (2024) highlight the role of transformed token magnitudes in ViT explanations. These scores are min-max normalized: $\hat{\alpha}_i = \frac{\alpha_i - \min_j \alpha_j}{\max_j \alpha_j - \min_j \alpha_j + \varepsilon}$, where $\varepsilon$ ensures numerical stability.

**Gradient Reweighting.** Each token's gradient is modulated by a scaling factor: Let $l \in \{1, \ldots, L\}$ denote the index of the current transformer block, and let $\mathcal{E} \subset \{1, \ldots, L\}$ be the set of early layers

(e.g., $\mathcal{E} = \{1, \ldots, k\}$). Define an indicator function:

$$\beta^{(l)} = \begin{cases} 1 & \text{if } l \in \mathcal{E} \quad \text{(early layer)} \\ 0 & \text{otherwise} \end{cases} \tag{1}$$

The final scaling factor for token $i$ at layer $l$ is then computed as: $s_i^{(l)} = \gamma_{\text{base}} + \lambda \cdot \left[(1 - \beta^{(l)}) \cdot \hat{\alpha}_i + \beta^{(l)} \cdot (1 - \hat{\alpha}_i)\right]$. And the FSGS-modulated gradient is: $\mathbf{g}_i^{(l),\text{FSGS}} = s_i^{(l)} \cdot \mathbf{g}_i^{(l)}$. Here, $\gamma_{\text{base}} \in (0, 1]$ ensures minimum gradient flow, while $\lambda$ controls the suppression strength for less important tokens. This reweighting selectively amplifies gradients linked to semantically meaningful content. FSGS is applied independently to the QKV projections, attention weights, and MLP layers, using module-specific hyperparameters $\lambda_{\text{qkv}}, \lambda_{\text{attn}}, \lambda_{\text{mlp}}$, allowing tailored control over each component. FSGS is implemented via backward hooks, imposes negligible overhead, and integrates seamlessly with iterative attack frameworks. By aligning perturbations with high-importance regions, it enhances the semantic coherence and transferability of adversarial examples across both homogeneous and heterogeneous architectures.

### 3.3 SPECTRAL SMOOTHNESS REGULARIZATION (SSR)

We propose *Spectral Smoothness Regularization (SSR)* to suppress high-frequency perturbation artifacts that hinder cross-architecture transferability. At each PGD iteration, SSR applies a differentiable Gaussian blur to the adversarial input, enforcing a low-pass constraint on the evolving perturbation: $\mathbf{x}_{adv}^{\text{blur}} = \mathcal{G}_\sigma(\mathbf{x} + \delta)$, where $\delta$ is the perturbation and $\mathcal{G}_\sigma(\cdot)$ denotes Gaussian blur with standard deviation $\sigma$. The motivation follows from both signal processing and adversarial transferability studies: high-frequency perturbations often overfit surrogate-specific features and fail to generalize (Tsipras et al., 2019; Yin et al., 2019), whereas lower-frequency structures better align with perceptually salient, transferable patterns. Unlike input diversity approaches (Xie et al., 2019), which randomize input transformations, SSR directly regularizes the spectral content of the perturbation itself. It also differs from smoothing-based defenses, since the blur is applied *during optimization*, shaping the perturbation rather than post-processing it. SSR is lightweight, parameter-free, and compatible with any gradient-based attack. In practice, it synergizes with FSGS by reducing high-frequency noise while preserving semantically aligned gradients, leading to stronger transferability in both black-box and cross-architecture scenarios.

### 3.4 MODULE-WISE GRADIENT MODULATION

Vision Transformers differ from CNNs not only in architecture but also in how features and gradients evolve with depth. Prior studies (Ming et al., 2024; Yosinski et al., 2014; Naseer et al., 2022) have shown that deeper transformer layers tend to encode more specialized, model-specific patterns (particularly in the attention maps) which can harm the transferability of adversarial perturbations. To address this, we introduce a *Module-wise gradient modulation* strategy that suppresses unstable gradients in deep attention layers and softly attenuates the gradient flow in all modules (Attention, QKV, MLP) based on their layer depth. Inspired by ATT (Ming et al., 2024), our approach consists of two key components:

**Selective Attention Truncation.** We truncate the gradients flowing through the *Attention module* for deep transformer blocks beyond a fixed threshold $l_{\text{cut}}$, by setting their attention gradients to zero: $\mathbf{g}_l^{\text{attn}} \leftarrow \mathbb{1}_{[l < l_{\text{cut}}]} \cdot \mathbf{g}_l^{\text{attn}}$. This effectively disables attention backpropagation in deeper layers, mitigating overfitting to model-specific global patterns.

**Module-Wise Gradient Weakening.** For all layers $l \in \{1, \ldots, L\}$ and modules $m \in \{\text{attn}, \text{qkv}, \text{mlp}\}$, we scale the gradients using a module-specific weakening factor $\omega^{(m)} \in (0, 1]$: $\mathbf{g}_l^{(m)} \leftarrow \omega^{(m)} \cdot \mathbf{g}_l^{(m)}$. This softly adjusts the contribution of each module based on its depth and functional role, before applying further refinement via FSGS. The weakening factors $\omega_m^{(l)}$ and the truncation layer threshold $l_{\text{cut}}$ are predefined based on empirical sensitivity, further hyperparameter sensitivity studies are provided in Appendix D.

All gradient weakening and truncation operations are applied via backward hooks before the application of FSGS. This ordering ensures that noisy gradients are first suppressed or removed, and only the semantically meaningful signals are preserved and amplified by FSGS. Importantly, our method remains fully differentiable and does not alter the model's forward pass, preserving compatibility

with any transformer backbone. The overall optimization algorithm and different hyper-parameters for training adversarial example are provided in Appendix B.

# 4 EXPERIMENTS

## 4.1 EXPERIMENT SETUP

**Dataset.** Following prior works (Wei et al., 2022; Zhang et al., 2023; Ming et al., 2024), we randomly selected 1,000 clean images from the ILSVRC2012 validation set (Russakovsky et al., 2015), ensuring that all surrogate models correctly classify each image with high confidence. This selection facilitates a consistent and fair evaluation of transferability between models.

**Models.** We employ four representative Vision Transformer models as surrogate architectures: ViT-B/16 (Dosovitskiy et al., 2021), PiT-B (Heo et al., 2021), CaiT-S24 (Touvron et al., 2021b), and Visformer-S (Chen et al., 2021). To assess cross-architecture generalization, we group evaluation into two categories: ViT-to-ViT and ViT-to-CNN transfer. For ViT-to-ViT, we use four unseen target ViTs: DeiT-B (Touvron et al., 2021a), TNT-S (Han et al., 2021), LeViT-256 (Graham et al., 2021), and ConViT-B (d'Ascoli et al., 2021). For ViT-to-CNN, we evaluate against four deep CNN models: Inception-v3 (Inc-v3), Inception-v4 (Inc-v4), Inception-ResNet-v2 (IncRes-v2), and ResNet-v2-152 (Res-v2) (Szegedy et al., 2016; 2017; He et al., 2016). Additionally, to evaluate robustness against adversarial defenses, we include three adversarially trained CNN models: Inc-v3-ens3, Inc-v4-ens4, and IncRes-v2-adv (Madry et al., 2018; Xu et al., 2022) and two adversarially trained ViTs: Swin-B (Mo et al., 2022) and XCiT-S (Debenedetti et al., 2023).

**Baselines.** We compare our method against a suite of strong baseline attacks. These include momentum- and variance-based methods such as MI-FGSM (MIM) (Dong et al., 2018b), VMI-FGSM (VMI) (Wang & He, 2021), and Skip Gradient Method (SGM) (Wu et al., 2020). We also include three state-of-the-art transformer-specific attacks: PNA (Wei et al., 2022), TGR (Zhang et al., 2023), and ATT (Ming et al., 2024), which incorporate attention structure or token-level heuristics into their gradient manipulation strategies. We also compare against diffusion-based attacks such as Diffattack (Chen et al., 2025).

**Evaluation Metrics.** We evaluate attack performance using the standard *Attack Success Rate* (ASR), defined as the proportion of adversarial examples that successfully fool the target model. Higher ASR ($\uparrow$) indicates stronger transferability.

**Parameter Settings.** All experiments use a maximum perturbation bound of $\epsilon = 16/255$, consistent with prior work (Zhang et al., 2023). The number of PGD iterations is set to $T = 10$, with a step size of $\eta = \epsilon/T = 1.6/255$. Momentum is used for stabilization with decay factor $\mu = 1.0$. Model- and method-specific hyperparameters follow their original settings unless otherwise stated. Input images are resized to $224 \times 224$, and the patch size for transformer models is fixed at $16 \times 16$. For spectral smoothness regularization, we apply Gaussian blur with fixed kernel size $(3 \times 3)$ and $\sigma = 0.5$. We set $\gamma_{\text{base}} = 0.5$. The weakening factors $\omega$, layer truncation threshold $l_{\text{cut}}$, and the adaptive scaling factor to $\lambda$ are tuned per model to balance the influence of QKV, Attention, and MLP gradients within the backward pass. The specific values of these hyperparameters are provided in Appendix B.

## 4.2 EVALUATING THE TRANSFERABILITY

We evaluate the black-box transferability of adversarial examples generated by TESSER across ViTs, CNNs, and adversarially defended CNNs. Table 1 shows results when attacking ViTs using ViT-based surrogates. TESSER achieves an average ASR of **83.2%**, outperforming the strongest baseline (ATT) by **+5.8%** and DiffAttack by **+12.2%**. On CNN targets, where ViT-based attacks typically degrade, TESSER maintains strong performance with **74.4%** ASR **+10.9%** higher than ATT. This indicates that our semantic and frequency-aware perturbations generalize beyond transformer-specific structures. TESSER's improvements are particularly notable on hybrid architectures like LeViT and ConViT, where both spatial alignment and cross-attention modeling are critical.

When facing adversarially trained CNNs (Table 3), TESSER achieves **53.55%** ASR, surpassing all baselines by a large margin. This suggests that TESSER generates perturbations that are not only transferable but also robust against strong defenses, an essential property for real-world attack scenarios. We also observe that the relative gains of TESSER vary across target types. For ViTs, the gains are moderate, likely because transformer-specific methods already perform reasonably well in this setting. However, the improvement is more pronounced on CNNs and defended CNNs, where

Table 1: The attack success rate (%) of various transfer-based attacks against eight ViT models and the average attack success rate (%) of all black-box models. The best results are highlighted in **bold**.

| Model | Attack | ViT-B/16 | PiT-B | CaiT-S/24 | Visformer-S | DeiT-B | TNT-S | LeViT-256 | ConViT-B | $Avg_{bb}$ |
|---|---|---|---|---|---|---|---|---|---|---|
| ViT-B/16 | MIM | **100.0*** | 34.5 | 64.1 | 36.5 | 64.3 | 50.2 | 33.8 | 66.0 | 49.9 |
| | VMI | **99.6*** | 48.8 | 74.4 | 49.5 | 73.0 | 64.8 | 50.3 | 75.9 | 62.4 |
| | SGM | **100.0*** | 36.9 | 77.1 | 40.1 | 77.9 | 61.6 | 40.2 | 78.4 | 58.9 |
| | PNA | **100.0*** | 45.2 | 78.6 | 47.7 | 78.6 | 62.8 | 47.1 | 79.5 | 62.8 |
| | TGR | **100.0*** | 49.5 | 85.0 | 53.8 | 85.6 | 73.1 | 56.5 | 85.4 | 69.8 |
| | DiffAttack | 96.3* | 60.1 | 70.4 | 63.3 | 75.4 | 71 | 57.5 | 74.36 | 71 |
| | FPR | **100*** | 37.7 | 77.9 | 40.0 | 77.2 | 74.3 | 42.1 | 78.0 | 65.9 |
| | FPR + GRA | 99.3* | 61.7 | 88.0 | 65.2 | 87.3 | 86.8 | 65.5 | 89.5 | 80.41 |
| | ATT | 99.9* | 57.5 | 90.3 | 63.9 | 90.8 | 82.0 | 66.8 | 90.8 | 77.4 |
| | Ours | **100*** | 61.7 | **94** | 68.3 | 92.5 | 85.6 | 72.2 | 91.4 | 83.2↑ |
| PiT-B | MIM | 24.7 | **100.0*** | 34.7 | 44.5 | 33.9 | 43.0 | 38.3 | 37.8 | 36.7 |
| | VMI | 38.9 | **99.7*** | 51.0 | 56.6 | 50.1 | 57.0 | 52.6 | 51.7 | 51.1 |
| | SGM | 41.8 | **100.0*** | 57.3 | 73.9 | 57.9 | 72.6 | 68.1 | 59.9 | 61.6 |
| | PNA | 47.9 | **100.0*** | 62.6 | 74.6 | 62.4 | 70.6 | 67.3 | 61.7 | 63.9 |
| | TGR | 60.3 | **100.0*** | 80.2 | 87.3 | 78.0 | 87.1 | 81.6 | 76.5 | 78.7 |
| | ATT | 69.6 | **100.0*** | 86.1 | 91.9 | 85.5 | 93.5 | 89.0 | 85.5 | 85.9 |
| | Ours | 74.9 | **100.0*** | 91.6 | 93.2 | 92.1 | 95 | 92.4 | 91.7 | 91.4↑ |
| CaiT-S/24 | MIM | 70.9 | 54.8 | **99.8*** | 55.1 | 90.2 | 76.4 | 54.8 | 88.5 | 70.1 |
| | VMI | 76.3 | 63.6 | **98.8*** | 67.3 | 88.5 | 82.3 | 67.0 | 88.1 | 76.2 |
| | SGM | 86.0 | 55.8 | **100.0*** | 68.2 | 97.7 | 91.1 | 74.9 | 96.7 | 81.5 |
| | PNA | 82.4 | 60.7 | **99.7*** | 67.7 | 95.7 | 86.9 | 67.1 | 94.0 | 79.2 |
| | TGR | 88.2 | 66.1 | **100.0*** | 75.4 | 98.8 | 92.8 | 74.7 | 97.9 | 84.8 |
| | ATT | 93.6 | 76.4 | **100.0*** | 85.9 | 99.4 | 96.9 | 87.4 | 98.8 | 91.2 |
| | Ours | 95.2 | 81.4 | **100*** | 90.3 | 99.6 | 97.5 | 90.7 | 98.9 | 94.2↑ |
| Visformer-S | MIM | 28.1 | 50.4 | 41.0 | **99.9*** | 36.9 | 51.9 | 49.4 | 39.6 | 42.5 |
| | VMI | 39.2 | 60.0 | 56.6 | **100.0*** | 54.1 | 62.8 | 59.1 | 54.4 | 55.2 |
| | SGM | 18.8 | 41.8 | 34.9 | **100.0*** | 31.2 | 52.1 | 52.7 | 29.5 | 37.3 |
| | PNA | 35.4 | 61.5 | 54.7 | **100.0*** | 51.0 | 66.3 | 64.5 | 50.7 | 54.9 |
| | TGR | 41.2 | 70.3 | 62.0 | **100.0*** | 59.5 | 74.7 | 74.8 | 56.2 | 62.7 |
| | ATT | 44.7 | 70.9 | 68.7 | **100.0*** | 66.4 | 78.8 | 80.9 | 58.4 | 67.0 |
| | Ours | **57.6** | **79.4** | **78.4** | **100.0*** | **75.9** | **83.2** | **85.3** | **69.6** | **78.7↑** |

ATT and TGR degrade significantly. This asymmetry suggests that our method is particularly effective at bridging the architectural gap between transformer and non-transformer models. Furthermore, TESSER's performance is more stable across all target types, showing lower variance than competing methods, which reinforces the robustness of our approach. Additional results and extended analysis are presented in Appendix C, in addition to a comparison with AutoAttack (Appendix F) and targeted attack evaluations (Appendix E).

We conducted additional experiments on robust ViT models trained via adversarial training with epsilon = 4, including Swin-B (Mo et al., 2022) and XCiT-S (Debenedetti et al., 2023). We compared TESSER against state-of-the-art attacks (PNA+PO, TGR+PO, and ATT+SPPO) using their optimal hyperparameters. As shown in Table 2, TESSER consistently achieves the highest ASR on both robust and corresponding standard ViT models, confirming its strong effectiveness even under adversarial defense settings. These results demonstrate that TESSER's transferability extends to robust ViTs, not just CNNs and hybrids.

## 4.3 ABLATION ON MODULE-WISE GRADIENT MODULATION

Table 2: The attack success rate (%) of various transfer-based attacks against robust ViTs. The best results are highlighted in **bold**).

| Model | Attack | Robust ViTs | | Normal ViTs | |
|---|---|---|---|---|---|
| | | Swin-B | Xcit-S | Swin-B | Xcit-S |
| ViT-B/16 | clean | 5.4 | 46.8 | 0.4 | 0.2 |
| | PNA+PO | 8.8 | 51.7 | 47.5 | 45.5 |
| | TGR+PO | 15.8 | 56.5 | 54.4 | 54.5 |
| | ATT+SPPO | 16.9 | 56.7 | 70.4 | 68.6 |
| | TESSER | 29.7↑ | 70.8↑ | 99.9↑ | 77.9↑ |
| PiT-B | PNA+PO | 9.2 | 51.8 | 67.0 | 71.2 |
| | TGR+PO | 17.9 | 58.2 | 77.3 | 80.7 |
| | ATT+SPPO | 18.7 | 58.3 | 90.4 | 92.8 |
| | TESSER | 31.9↑ | 71.6↑ | 100↑ | 95.4↑ |

To understand the individual and combined contributions of our gradient modulation strategy across different transformer modules, we conduct an ablation study by selectively applying Feature-Sensitive Gradient Scaling to the Attention, QKV, and MLP components. Table 4 presents the attack success rates (ASR) on ViT-based models, CNNs, and defended CNNs under different configurations. When FSGS is applied to a single module, the Attention pathway contributes the most to transferability, particularly for ViTs, achieving an ASR of 80.1%. MLP-only and QKV-only configurations also yield strong improvements over the baseline, with notable gains on CNNs and defended mod-

els. Combining any two modules improves performance further, especially when including MLP, which significantly boosts ASR against robust models. The best results are obtained when FSGS is jointly applied to all three modules, yielding an ASR of 86.88% on ViTs and 53.55% on defended CNNs. These results confirm that our gradient modulation strategy is most effective when applied in a comprehensive and module-aware manner.

Table 3: The attack success rate (%) of various transfer-based attacks against four undefended CNN models and three defended CNN models and the average attack success rate (%) of all black-box models. The best results are highlighted in **bold**.

| Model | Attack | Inc-v3 | Inc-v4 | IncRes-v2 | Res-v2 | Inc-v3ens3 | Inc-v3ens4 | IncRes-v2adv | Avg$_{bb}$ |
|---|---|---|---|---|---|---|---|---|---|
| ViT-B/16 | MIM | 31.7 | 28.6 | 26.1 | 29.4 | 22.3 | 19.8 | 16.5 | 24.9 |
| | VMI | 43.1 | 41.6 | 37.9 | 42.6 | 31.4 | 30.6 | 25.0 | 36.0 |
| | SGM | 31.5 | 27.7 | 23.8 | 28.2 | 20.8 | 18.0 | 14.3 | 23.5 |
| | PNA | 42.7 | 37.5 | 35.3 | 39.5 | 29.0 | 27.3 | 22.6 | 33.4 |
| | TGR | 47.5 | 42.3 | 37.6 | 43.3 | 31.5 | 30.8 | 25.6 | 36.9 |
| | DiffAttack | 55.9 | 53.4 | 52.1 | 56.8 | 45.8 | 48.7 | 41.5 | 50.6 |
| | FPR | 30.5 | 26.1 | 23.0 | 25.8 | 17.7 | 15.1 | 11.0 | 21.3 |
| | FPR + GRA | 52.9 | 48.0 | 45.4 | 47.9 | 43.4 | 42.3 | 33.5 | 44.7 |
| | ATT | 53.3 | 49.0 | 45.4 | 51.5 | 38.1 | 36.7 | 33.1 | 43.9 |
| | Ours | **63.4** | **59.6** | **54.4** | **57.7** | **48.6** | **49** | **42.3** | **53.6**↑ |
| PiT-B | MIM | 36.3 | 34.8 | 27.4 | 29.6 | 19.0 | 18.3 | 14.1 | 25.6 |
| | VMI | 47.3 | 45.4 | 40.7 | 43.4 | 35.9 | 34.4 | 29.7 | 39.5 |
| | SGM | 50.6 | 45.4 | 38.4 | 41.9 | 25.6 | 20.8 | 16.7 | 34.2 |
| | PNA | 59.3 | 56.3 | 49.8 | 53.0 | 33.3 | 32.0 | 25.5 | 44.2 |
| | TGR | 72.1 | 69.8 | 65.1 | 64.8 | 43.6 | 41.5 | 32.8 | 55.7 |
| | ATT | 80.4 | 75.3 | 72.7 | 72.9 | 52.5 | 50.6 | 41.0 | 63.6 |
| | Ours | **87.2** | **87.5** | **78.4** | **80** | **61** | **61.3** | **48.9** | **72**↑ |
| CaiT-S/24 | MIM | 48.4 | 42.9 | 39.5 | 43.8 | 30.8 | 27.6 | 23.3 | 36.6 |
| | VMI | 58.5 | 50.9 | 48.2 | 52.0 | 38.1 | 36.1 | 30.1 | 44.8 |
| | SGM | 53.5 | 45.9 | 40.2 | 45.9 | 30.8 | 28.5 | 21.0 | 38.0 |
| | PNA | 57.2 | 51.8 | 47.7 | 51.6 | 38.4 | 36.2 | 30.1 | 44.7 |
| | TGR | 60.3 | 52.9 | 49.3 | 53.4 | 39.6 | 37.0 | 31.8 | 46.3 |
| | ATT | 73.9 | 66.0 | 66.3 | 66.4 | 54.6 | 52.1 | 43.9 | 60.5 |
| | Ours | **79.2** | **71.9** | **72** | **72.4** | **57.9** | **57.5** | **49.2** | **65.7**↑ |
| Visformer-S | MIM | 44.5 | 42.5 | 36.6 | 39.6 | 24.4 | 20.5 | 16.6 | 32.1 |
| | VMI | 54.6 | 53.2 | 48.5 | 52.2 | 33.0 | 32.0 | 22.2 | 42.2 |
| | SGM | 43.2 | 41.1 | 29.6 | 35.7 | 16.1 | 13.0 | 8.2 | 26.7 |
| | PNA | 55.9 | 54.6 | 46.0 | 51.7 | 29.3 | 26.2 | 21.1 | 40.7 |
| | TGR | 65.9 | 66.8 | 55.3 | 60.9 | 36.0 | 32.5 | 23.3 | 48.7 |
| | ATT | 80.9 | 81.2 | 70.5 | 75.7 | 50.1 | 41.3 | 32.0 | 61.7 |
| | Ours | **84.2** | **84.6** | **77.3** | **80.6** | **64.6** | **57.4** | **45** | **70.5**↑ |

### 4.4 QUALITATIVE COMPARISON: PERTURBATION SEMANTICS

Table 4: The average attack success rate (%) against ViTs, CNNs, and defended CNNs by our method with different module settings.

| Attn | QKV | MLP | ViTs | CNNs | Def-CNNs |
|---|---|---|---|---|---|
| – | – | – | 49.9 | 29.1 | 19.3 |
| ✓ | – | – | 80.1 | 61.1 | 36.1 |
| – | ✓ | – | 72.72 | 54.1 | 29.5 |
| – | – | ✓ | 71.87 | 59 | 36 |
| ✓ | ✓ | – | 78.43 | 55.4 | 30.3 |
| ✓ | – | ✓ | 83.32 | 70.9 | 52 |
| – | ✓ | ✓ | 81.21 | 66.3 | 39.9 |
| ✓ | ✓ | ✓ | 86.88 | 74.4 | 53.55 |

We visualize adversarial examples generated by ATT (Ming et al., 2024) and our proposed FSGS to examine the semantic alignment of perturbations. Each case includes the clean image, the adversarial example, and a Grad-CAM heatmap computed from the adversarial prediction of a black-box model. As shown in Figure 2, FSGS perturbations remain spatially aligned with semantically salient regions (e.g., object parts or discriminative textures), even when the model misclassifies the input. In contrast, ATT tends to spread noise across the image without clear semantic focus. These results validate our central assumption: token activation norms correlate with semantic importance, and preserving gradients from high-norm tokens guides perturbations toward class-relevant features. This not only improves interpretability but also enhances transferability across architectures.

**Evaluating TESSER Perceptual Stealthiness.** While SSR reduces high-frequency components, it operates only during the perturbation generation phase. The final adversarial image is obtained by subtracting the noise and clamping the result. Thus, no direct blurring is applied to the image itself, preserving spatial clarity. To objectively assess perceptual visibility, we provide a quantitative

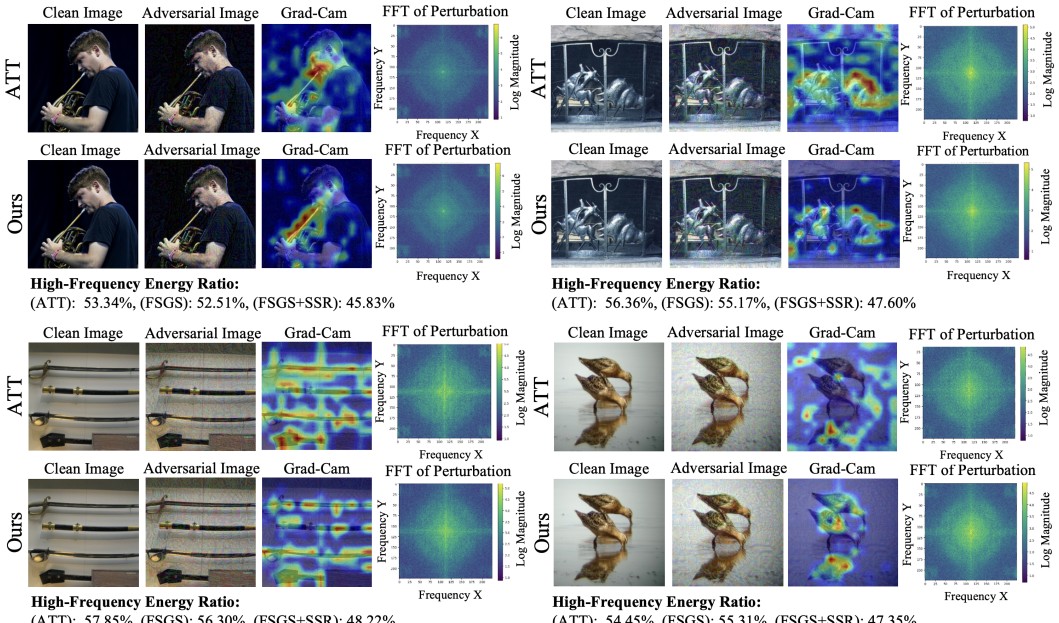

Figure 2: **Qualitative and frequency-domain comparison between ATT and our method (FSGS and FSGS + SSR).** Each row shows clean images, adversarial examples when using FSGS, Grad-CAM (guided by the adversarial label) overlays, and FFT log-magnitude spectra when using SSR. Our method produces perturbations that better align with semantically relevant regions and exhibit smoother frequency profiles. Further results and analysis are provided in Appendix D.

comparison using LPIPS, SSIM, and PSNR across TESSER and transfer-based attacks such as ATT and TGR. As shown in Table 5, TESSER achieves significantly higher imperceptibility, with 50% reduction in LPIPS, 33% improvement in SSIM, and +5 dB increase in PSNR, demonstrating strong stealthiness without sacrificing effectiveness.

## 4.5 SPECTRAL SMOOTHNESS EVALUATION VIA FREQUENCY-DOMAIN ANALYSIS

To quantitatively assess the effect of Spectral Smoothness Regularization (SSR), we conduct a frequency-domain analysis of the generated perturbations. Specifically, we compute the 2D Fast Fourier Transform (FFT) of each perturbation and evaluate the *high-frequency energy ratio*, defined as the proportion of energy outside the central low-frequency band in the log-magnitude spectrum (as shown in Figure 2). Given a perturbation $\delta \in \mathbb{R}^{3 \times H \times W}$, we compute its FFT, shift the spectrum to center the low frequencies, and apply a radial mask to isolate high-frequency components.

Table 5: Stealth Evaluation of Transfer-Based Attacks.

| Metric | TGR | ATT | TESSER |
|---|---|---|---|
| LPIPS $\downarrow$ | 0.35 | 0.42 | **0.21** |
| SSIM $\uparrow$ | 0.66 | 0.57 | **0.77** |
| PSNR $\uparrow$ | 22.23 dB | 19.70 dB | **25.04 dB** |

This experiment is repeated on a batch of adversarial samples to compare the spectral concentration of different attack variants. Our results demonstrate that SSR substantially reduces the high-frequency energy of perturbations. Across examples, ATT shows the highest high-frequency ratios (e.g., 53-56%), while FSGS reduces this moderately ($\sim$52–55%). When combined with SSR, the high-frequency ratio drops further (to $\sim$45–47%), indicating smoother and more transferable perturbations. This confirms that SSR encourages low-frequency perturbation structure, complementing the token-aware gradient modulation of FSGS.

## 5 CONCLUSION

We proposed TESSER, a unified adversarial attack framework designed to improve transferability across diverse model architectures. By integrating Feature-Sensitive Gradient Scaling (FSGS) and Spectral Smoothness Regularization (SSR), TESSER guides adversarial gradients through semantically meaningful token activations and enforces smooth, low-frequency perturbation structures. Combined with layer and module-wise gradient modulation, our method effectively mitigates overfitting to model-specific representations and enhances generalization to unseen targets. Experimental results across a wide range of ViTs, hybrid models, and CNNs demonstrate that TESSER consistently outperforms state-of-the-art transfer attacks in both accuracy degradation and optimization efficiency.

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

APPENDIX

# A  THEORETICAL JUSTIFICATION OF FEATURE-SENSITIVE GRADIENT SCALING (FSGS)

We provide a formal argument to support the hypothesis that modulating gradients based on token-level activation norms enhances adversarial transferability. Our analysis is grounded in the relationship between semantic informativeness and gradient alignment across models.

## A.1  PRELIMINARIES

Let $f : \mathbb{R}^d \to \mathbb{R}^K$ be a surrogate classifier and $f' : \mathbb{R}^d \to \mathbb{R}^K$ a target (black-box) classifier. An adversarial perturbation $\delta \in \mathbb{R}^d$ is added to input $\mathbf{x}$ such that $\|\delta\|_\infty \leq \epsilon$ and $f(\mathbf{x} + \delta) \neq y$.

Assume $\mathbf{x}$ is decomposed into $T$ tokens with embeddings $\mathbf{z}_1, \ldots, \mathbf{z}_T \in \mathbb{R}^D$. Denote the loss gradients w.r.t. each token as $\mathbf{g}_i = \nabla_{\mathbf{z}_i} \mathcal{L}(f(\mathbf{x}), y)$, and similarly for $f'$.

## A.2  SEMANTIC TOKENS AND GRADIENT ALIGNMENT

Let $S_{\text{sem}} \subseteq \{1, \ldots, T\}$ be the set of semantically informative tokens (e.g., foreground object, discriminative parts). Let $S_{\text{bg}}$ be its complement (background or irrelevant tokens).

We define the inter-model gradient alignment at token $i$ as:
$$\text{Align}_i = \cos \theta_i = \frac{\langle \nabla_{\mathbf{z}_i} \mathcal{L}_f, \nabla_{\mathbf{z}_i} \mathcal{L}_{f'} \rangle}{\|\nabla_{\mathbf{z}_i} \mathcal{L}_f\| \cdot \|\nabla_{\mathbf{z}_i} \mathcal{L}_{f'}\|}$$

**Assumption 1.** *Gradients at semantically important tokens exhibit higher cross-model alignment:*
$$\mathbb{E}_{i \in S_{sem}}[Align_i] > \mathbb{E}_{i \in S_{bg}}[Align_i]$$

This is supported by empirical findings in model interpretability (Abnar & Zuidema, 2020; Lin & Parikh, 2016; Raghu et al., 2021) and our own Grad-CAM visualizations (see Section 4.4).

## A.3  FEATURE-SENSITIVE GRADIENT SCALING (FSGS)

FSGS assigns a scaling factor $s_i$ to each token based on its activation norm $\alpha_i = \|\mathbf{z}_i\|_2$:
$$s_i = \gamma_{\text{base}} + \lambda(1 - \hat{\alpha}_i), \quad \hat{\alpha}_i = \frac{\alpha_i - \min_j \alpha_j}{\max_j \alpha_j - \min_j \alpha_j + \varepsilon}$$

Tokens with high $\alpha_i$ (assumed to lie in $S_{\text{sem}}$) receive larger gradients, while low-importance tokens are suppressed.

**Theorem 1** (FSGS Improves Expected Gradient Alignment). *Let $G = \sum_{i=1}^{T} \mathbf{g}_i$ be the unscaled gradient and $G_{FSGS} = \sum_{i=1}^{T} s_i \cdot \mathbf{g}_i$ the FSGS-scaled gradient. Under Assumption 1, the cosine alignment between $G_{FSGS}$ and the target model's gradient $G'$ satisfies:*
$$\cos \theta(G_{FSGS}, G') > \cos \theta(G, G')$$

*Sketch.* We decompose the total gradient into two subsets:
$$G = \sum_{i \in S_{\text{sem}}} \mathbf{g}_i + \sum_{i \in S_{\text{bg}}} \mathbf{g}_i$$

FSGS scales $i \in S_{\text{sem}}$ by higher $s_i$ than those in $S_{\text{bg}}$, thus:
$$G_{\text{FSGS}} = \sum_{i \in S_{\text{sem}}} s_i \mathbf{g}_i + \sum_{i \in S_{\text{bg}}} s_i \mathbf{g}_i$$

Since $\mathbb{E}_{i \in S_{\text{sem}}}[\text{Align}_i] > \mathbb{E}_{i \in S_{\text{bg}}}[\text{Align}_i]$, amplifying contributions from $S_{\text{sem}}$ increases the expected alignment between $G_{\text{FSGS}}$ and $G'$. Therefore:

$\cos \theta(G_{\text{FSGS}}, G') > \cos \theta(G, G')$   (by Jensen's inequality over positively weighted aligned vectors)

$\square$

## A.4 IMPLICATION

Theorem 1 provides theoretical support for the design of FSGS: boosting gradients from semantically salient tokens leads to improved alignment with gradients from unseen models, thereby enhancing adversarial transferability. This also explains the empirical advantage of FSGS in black-box settings, particularly when transferring from ViTs to CNNs or hybrid models.

# B THE OVERALL FRAMEWORK OF OPTIMIZATION ALGORITHM

## B.1 ALGORITHM

We now present the full optimization framework used to generate adversarial examples in our method. Our algorithm builds on the momentum-based PGD attack (Dong et al., 2018a), and integrates three coordinated components: (1) *Module and Layer-Wise Gradient Modulation* to adjust the contribution of each transformer layer and suppress noisy deep-layer gradients, (2) *Feature-Sensitive Gradient Scaling (FSGS)* to selectively enhance semantically important token gradients, and (3) *Spectral Smoothness Regularization (SSR)* to constrain the perturbation's frequency content.

Let $\mathbf{x} \in \mathbb{R}^{C \times H \times W}$ be a clean input, $y \in \{1, \dots, K\}$ its ground-truth label, and $f$ the surrogate model. We seek a perturbation $\delta$ satisfying $\|\delta\|_\infty \leq \epsilon$, such that the adversarial input $\mathbf{x}^{\text{adv}} = \mathbf{x} + \delta$ misleads $f$ and transfers effectively to other black-box models.

OPTIMIZATION PROCEDURE

The perturbation is optimized over $T$ steps using projected gradient descent with momentum. At each step $t \in \{1, \dots, T\}$, the perturbed input is smoothed using a differentiable Gaussian blur operator:

$$\mathbf{x}^{(t)} = \mathcal{G}_\sigma(\mathbf{x} + \delta^{(t-1)})$$

where $\mathcal{G}_\sigma(\cdot)$ denotes Gaussian blurring with standard deviation $\sigma$, enforcing low-frequency spectral structure (SSR).

Next, the gradient of the loss is computed:

$$\mathbf{g}^{(t)} = \nabla_{\mathbf{x}} \mathcal{L}(f(\mathbf{x}^{(t)}), y)$$

This gradient is intercepted via backward hooks at key ViT modules (Attention, QKV, MLP). For each transformer block $l$, the following sequence is applied to each module:

1. **Module-wise Weakening:** The gradient $\mathbf{g}^{(l)}$ for each module is first scaled using a module-specific weakening factor $\omega^{(l)} \in (0, 1]$ (e.g., $\omega_{\text{attn}}^{(l)}, \omega_{\text{qkv}}^{(l)}, \omega_{\text{mlp}}^{(l)}$). This captures prior knowledge about the sensitivity of each module.

2. **Layer-wise Modulation:** The weakened attention gradient is then further modulated by a layer-specific coefficient $\tau_l \in [0, 1]$, which reduces the influence of deeper transformer layers:

$$\mathbf{g}^{(l)} \leftarrow \tau_l \cdot (\omega^{(l)} \cdot \mathbf{g}^{(l)})$$

3. **Feature-Sensitive Gradient Scaling (FSGS):**

   A layer-aware gradient modulation mechanism that scales gradients based on token-wise importance. FSGS promotes perturbation alignment with semantically salient features while suppressing low-level, architecture-specific signals that degrade cross-model transferability. Let $\mathbf{Z} \in \mathbb{R}^{T \times D}$ be the token embedding matrix at a given transformer block. We define the raw importance score of token $i$ as: $\alpha_i = \|\mathbf{z}_i\|_2$. The importance scores are min-max normalized across tokens: $\hat{\alpha}_i = \frac{\alpha_i - \min_j \alpha_j}{\max_j \alpha_j - \min_j \alpha_j + \varepsilon}$, where $\varepsilon$ is a small constant to avoid division by zero.

   Let $l \in \{1, \dots, L\}$ denote the index of the current transformer block, and let $\mathcal{E} \subset \{1, \dots, L\}$ be the set of early layers (e.g., $\mathcal{E} = \{1, \dots, k\}$). Define an indicator function:

$$\beta^{(l)} = \begin{cases} 1 & \text{if } l \in \mathcal{E} \quad \text{(early layer)} \\ 0 & \text{otherwise} \end{cases} \tag{2}$$

The final scaling factor for token $i$ at layer $l$ is then computed as: $s_i^{(l)} = \gamma_{\text{base}} + \lambda \cdot \left[(1 - \beta^{(l)}) \cdot \hat{\alpha}_i + \beta^{(l)} \cdot (1 - \hat{\alpha}_i)\right]$. And the FSGS-modulated gradient is: $\mathbf{g}_i^{(l),\text{FSGS}} = s_i^{(l)} \cdot \mathbf{g}_i^{(l)}$

All module gradients are aggregated to form the total input gradient $\mathbf{g}^{(t)}$, and momentum is applied:

$$\mathbf{m}^{(t)} = \mu \cdot \mathbf{m}^{(t-1)} + \frac{\mathbf{g}^{(t)}}{\|\mathbf{g}^{(t)}\|_1}$$

The perturbation is updated and projected onto the $\ell_\infty$-norm ball:

$$\delta^{(t)} = \text{Clip}_\epsilon\left(\delta^{(t-1)} + \eta \cdot \text{sign}(\mathbf{m}^{(t)})\right)$$

**Description.** Algorithm 1 summarizes our full optimization loop. The key innovation lies in the sequential application of module-wise weakening, layer-wise modulation, and semantic-aware scaling via FSGS. All components are implemented via backward hooks, ensuring compatibility with any transformer-based model.

**Hyper-parameters.** Table 6 summarizes the model-specific hyperparameters used in TESSER for each architecture. These include module-wise weakening factors ($\omega^{(\cdot)}$), FSGS scaling parameters ($\lambda$.), spectral smoothness regularization strength ($\sigma$), attention truncation depth ($l_{\text{cut}}$), base gradient scaling ($\gamma_{\text{base}}$), as well as optimization parameters: momentum decay ($\mu$) and step size ($\eta$). Values are carefully selected to balance gradient modulation and attack stability per architecture.

## B.2 Computational Cost

To evaluate the efficiency of our proposed method, we report the average time (in seconds) required to generate a single adversarial example using FSGS, FSGS+SSR, and the ATT across different models. As shown in Table 7, our methods incur minimal overhead compared to ATT, with only a slight increase when applying SSR. In particular, even in deeper architectures like CaiT-S/24, FSGS+SSR remains significantly more efficient than ATT.

We provide the detailed environment configuration used for all evaluations. All experiments were conducted on NVIDIA Tesla T4 GPUs hosted on Google Colab. We present the key software dependencies and their corresponding versions:

- Python: 3.11.12
- PyTorch: 2.6.0
- Torchvision: 0.21.0
- NumPy: 2.0.2
- Pillow: 11.2.1
- Timm: 1.0.15
- SciPy: 1.15.3

## C Additional Experiments

### C.1 Quantitative Analysis for SSR

To evaluate the effect of spectral regularization strength, we conduct an ablation study by varying the Gaussian blur standard deviation $\sigma$ used in Spectral Smoothness Regularization (SSR). Table 8 reports the average attack success rates (ASR) on ViTs, CNNs, and defended CNNs for $\sigma \in \{0.5, 0.6, 0.7, 0.8\}$ across all surrogate models.

We observe a consistent trend: increasing $\sigma$ improves transferability to CNNs and defended CNNs, while slightly reducing ASR on ViTs. This trade-off reflects the role of SSR in suppressing high-frequency architecture-specific noise: improving cross-architecture generalization but marginally

**Algorithm 1: TESSER: Transfer-Enhancing Adversarial Optimization from Vision Transformers via Spectral and Semantic Regularization**

---

**Input:** Input image $\mathbf{x}$, label $y$, model $f$,
Steps $T$, step size $\eta$, perturbation bound $\epsilon$,
Gaussian blur $\mathcal{G}_\sigma$, momentum $\mu$,
Base scale $\gamma_{\text{base}}$,
Module-specific FSGS strengths $\lambda_{\text{qkv}}, \lambda_{\text{attn}}, \lambda_{\text{mlp}}$,
Early-layer set $\mathcal{E}$, attention cutoff layer $l_{\text{cut}}$,
Module weakening factors $\omega^{(l)}$,
SSR loss function $\mathcal{L}_{\text{SSR}}$
**Output:** Adversarial example $\mathbf{x}^{\text{adv}}$
**Initialize:** $\delta^{(0)} = 0$, $\mathbf{m}^{(0)} = 0$
**for** $t = 1$ **to** $T$ **do**

    **1. Apply SSR:**
    $\mathbf{x}^{(t)} = \mathcal{G}_\sigma(\mathbf{x} + \delta^{(t-1)})$
    **2. Forward pass and compute classification loss:**
    $\mathcal{L}_{\text{cls}}^{(t)} = \mathcal{L}(f(\mathbf{x}^{(t)}), y)$
    **3. Backward pass with hooks at QKV, Attention, and MLP modules:**
    **foreach** *block* $l \in \{1, \ldots, L\}$ **do**

        **foreach** *module* $m \in \{qkv, attn, mlp\}$ **do**

            **3.1 Extract token features and gradients:**
            $\mathbf{Z}^{(l,m)} = [\mathbf{z}_1^{(l,m)}, \ldots, \mathbf{z}_T^{(l,m)}]$
            $\mathbf{G}^{(l,m)} = [\mathbf{g}_1^{(l,m)}, \ldots, \mathbf{g}_T^{(l,m)}]$
            **3.2 Compute token importance:**
            $\alpha_i = \|\mathbf{z}_i^{(l,m)}\|_2, \quad \hat{\alpha}_i = \frac{\alpha_i - \min_j \alpha_j}{\max_j \alpha_j - \min_j \alpha_j + \varepsilon}$
            **3.3 Apply module-wise weakening:**
            $\mathbf{G}^{(l,m)} \leftarrow \omega^{(l)} \cdot \mathbf{G}^{(l,m)}$
            **3.4 Selective Attention Truncation (only if $m = $ attn):**
            **if** $l \geq l_{cut}$ **then**
                $\mathbf{G}^{(l,\text{attn})} \leftarrow 0$
            **end**
            **3.5 Compute FSGS scaling:**
            **foreach** *token* $i \in \{1, \ldots, T\}$ **do**

                **if** $l \in \mathcal{E}$ **then**
                    $s_i = \gamma_{\text{base}} + \lambda_m \cdot (1 - \hat{\alpha}_i)$
                **end**
                **else**
                    $s_i = \gamma_{\text{base}} + \lambda_m \cdot \hat{\alpha}_i$
                **end**
                $\mathbf{g}_i^{(l,m)} \leftarrow s_i \cdot \mathbf{g}_i^{(l,m)}$
            **end**
        **end**
    **end**
    **4. Aggregate gradients across all modules:**
    $\mathbf{g}^{(t)} = \sum_{l,m} \text{Aggregate}(\mathbf{G}^{(l,m)})$
    **5. Momentum update:**
    $\mathbf{m}^{(t)} = \mu \cdot \mathbf{m}^{(t-1)} + \frac{\mathbf{g}^{(t)}}{\|\mathbf{g}^{(t)}\|_1}$
    **6. Perturbation update with projection:**
    $\delta^{(t)} = \text{Clip}_\epsilon(\delta^{(t-1)} + \eta \cdot \text{sign}(\mathbf{m}^{(t)}))$
**end**
**return** $\mathbf{x}^{\text{adv}} = \mathbf{x} + \delta^{(T)}$

---

Table 6: Model-specific hyperparameter settings used for TESSER. $\omega^{(\cdot)}$ denotes the weakening factor for each module, $\lambda.$ is the FSGS scaling parameter, $\sigma$ controls the strength of spectral regularization, $l_{\text{cut}}$ is the attention truncation depth, $\gamma_{\text{base}}$ is the minimum gradient scaling factor, $\mu$ is the momentum decay used in PGD, and $\eta$ is the step size for perturbation update.

| Hyperparameter | ViT-B/16 | PiT-B | CaiT-S/24 | Visformer-S |
|---|---|---|---|---|
| $\omega^{(\text{attn})}$ | 0.45 | 0.25 | 0.3 | 0.4 |
| $\omega^{(\text{qkv})}$ | 0.5 | 0.5 | 1.0 | 0.8 |
| $\omega^{(\text{mlp})}$ | 0.7 | 0.7 | 0.6 | 0.5 |
| $\lambda_{\text{attn}}$ | 0.4 | 0.45 | 0.5 | 0.45 |
| $\lambda_{\text{qkv}}$ | 0.5 | 0.5 | 0.5 | 0.5 |
| $\lambda_{\text{mlp}}$ | 0.55 | 0.55 | 0.65 | 0.6 |
| $\sigma$ (SSR) | 0.5 | 0.7 | 0.7 | 0.7 |
| $l_{\text{cut}}$ | 10 | 9 | 4 | 8 |
| $\gamma_{\text{base}}$ | 0.5 | 0.5 | 0.5 | 0.5 |
| $\mu$ | 1.0 | 1.0 | 1.0 | 1.0 |
| $\eta$ | 1.6/255 | 1.6/255 | 1.6/255 | 1.6/255 |

Table 7: Computational cost (in seconds) for generating a single adversarial example across different models and methods. FSGS refers to our feature-sensitive gradient scaling, SSR refers to spectral smoothness regularization, and ATT denotes state of the art.

| Model | FSGS | FSGS + SSR | ATT (Ming et al., 2024) |
|---|---|---|---|
| ViT-B/16 | 0.5 | 0.52 | 0.93 |
| PiT-B | 0.54 | 0.6 | 1.05 |
| CaiT-S/24 | 1.24 | 1.27 | 1.88 |
| Visformer-S | 0.35 | 0.38 | 1.14 |

weakening model-specific alignment. For instance, in ViT-B/16, increasing $\sigma$ from 0.5 to 0.8 decreases ViT ASR from 83.21% to 81.21%, but improves CNN ASR from 58.77% to 61.82% and defended CNN ASR from 46.63% to 52.33%. A similar pattern is observed in PiT-B and CaiT-S/24.

Notably, the improvement on defended CNNs is particularly pronounced. For Visformer-S, the ASR on defended models improves from 46.96% at $\sigma = 0.5$ to 61.5% at $\sigma = 0.8$, a gain of over 14%. These results confirm that SSR strengthens black-box transferability and robustness by encouraging low-frequency perturbations that are less dependent on the surrogate model's internal architecture.

In practice, setting $\sigma$ between 0.6 and 0.8 offers a favorable trade-off, preserving sufficient ViT ASR while achieving substantial improvements on CNNs and defended models. This ablation supports the effectiveness of spectral regularization and its role in enhancing generalization under diverse adversarial settings.

## C.2 COMPARISON OF ATTACK EFFICIENCY WHEN USING INPUT DIVERSITY TECHNIQUE

To further assess the effectiveness and generality of our proposed TESSER framework, we evaluate its performance when combined with an input diversity enhancement strategy, specifically PatchOut (PO) (Wei et al., 2022). This technique introduces random masking during inference to improve the robustness and transferability of adversarial perturbations.

Table 9 presents the average Attack Success Rate (ASR) of different attack methods augmented with PO, tested across ViTs, CNNs, and defended CNNs. The experiments span four representative surrogate models: ViT-B/16, CaiT-S/24, PiT-B, and Visformer-S.

Across all surrogate models and evaluation categories, TESSER+PO consistently achieves the highest ASR. For example, using PiT-B as the surrogate, TESSER+PO achieves an ASR of **94.83%** on ViTs, **87.7%** on CNNs, and **61.43%** on defended CNNs, representing improvements of more than **+10%** over the strongest baseline ATT+PO. Similar trends are observed with the other surrogate models,

Table 8: Average attack success rate (ASR) (%) against ViTs, CNNs, and defended CNNs across varying Gaussian blur strength $\sigma$. Increasing $\sigma$ generally improves transferability to CNNs and defended models by enforcing low-frequency perturbations, while slightly reducing white-box ASR on ViTs.

| Model | $\sigma$ | ViTs | CNNs | Def-CNNs | Model | $\sigma$ | ViTs | CNNs | Def-CNNs |
|-------|----------|------|------|----------|-------|----------|------|------|----------|
| ViT-B/16 | 0.5 | 83.21 | 58.77 | 46.63 | CaiT-S/24 | 0.5 | 94.82 | 68.12 | 45.6 |
|  | 0.6 | 83.12 | 61.85 | 49.86 |  | 0.6 | 94.57 | 71.9 | 51.76 |
|  | 0.7 | 81.95 | 61.62 | 51.13 |  | 0.7 | 94.2 | 73.87 | 54.86 |
|  | 0.8 | 81.21 | 61.82 | 52.33 |  | 0.8 | 93.82 | 73.55 | 57.06 |
| PiT-B | 0.5 | 90.3 | 80.85 | 49 | Visformer-S | 0.5 | 75.93 | 76.77 | 46.96 |
|  | 0.6 | 91.63 | 83.4 | 54.9 |  | 0.6 | 78.47 | 80.45 | 53.9 |
|  | 0.7 | 91.36 | 83.27 | 57.06 |  | 0.7 | 78.67 | 81.67 | 58.46 |
|  | 0.8 | 90.02 | 83.45 | 58.6 |  | 0.8 | 83.33 | 81.22 | 61.5 |

Table 9: The average attack success rate (%) against ViTs, CNNs, and defended CNNs by various transfer-based attacks with input diversity enhancement strategy. The best results are highlighted in bold. "PO" denotes PatchOut (Wei et al., 2022).

| Model | Attack | ViTs | CNNs | Def-CNNs | Model | Attack | ViTs | CNNs | Def-CNNs |
|-------|--------|------|------|----------|-------|--------|------|------|----------|
| ViT-B/16 | MIM+PO | 61.3 | 31.3 | 21.7 | CaiT-S/24 | MIM+PO | 70.3 | 44.0 | 29.3 |
|  | VMI+PO | 69.1 | 42.8 | 30.9 |  | VMI+PO | 76.8 | 57.8 | 38.4 |
|  | SGM+PO | 64.8 | 29.2 | 18.9 |  | SGM+PO | 85.1 | 49.2 | 29.3 |
|  | PNA+PO | 70.8 | 42.6 | 29.9 |  | PNA+PO | 81.6 | 56.6 | 39.3 |
|  | TGR+PO | 76.0 | 46.7 | 33.3 |  | TGR+PO | 88.8 | 60.5 | 40.5 |
|  | ATT+PO | 77.1 | 51.7 | 37.1 |  | ATT+PO | 91.1 | 71.9 | 54.3 |
|  | **Ours+PO** | **85.18↑** | **64.17↑** | **52.16↑** |  | **Ours+PO** | **91.15↑** | **72.9↑** | **56.46↑** |
| PiT-B | MIM+PO | 47.3 | 32.5 | 17.5 | Visformer-S | MIM+PO | 54.9 | 45.7 | 23.4 |
|  | VMI+PO | 59.5 | 46.2 | 35.8 |  | VMI+PO | 64.8 | 56.6 | 32.6 |
|  | SGM+PO | 70.0 | 45.6 | 21.3 |  | SGM+PO | 51.6 | 44.3 | 15.0 |
|  | PNA+PO | 73.1 | 57.8 | 32.7 |  | PNA+PO | 68.8 | 61.8 | 32.3 |
|  | TGR+PO | 82.3 | 68.9 | 41.3 |  | TGR+PO | 70.4 | 64.3 | 33.5 |
|  | ATT+PO | 84.2 | 75.2 | 48.4 |  | ATT+PO | 70.5 | 79.3 | 44.5 |
|  | **Ours+PO** | **94.83↑** | **87.7↑** | **61.43↑** |  | **Ours+PO** | **84.42↑** | **79.4↑** | **58.06↑** |

including CaiT-S/24 and ViT-B/16, where TESSER+PO continues to outperform baselines by wide margins.

These results demonstrate two key insights: (1) TESSER is orthogonal to input diversity methods, as its performance improves further when used with PO, and (2) our gradient modulation and spectral regularization strategies remain effective under randomized input transformations, indicating strong generalization.

In particular, on defended CNNs, traditionally difficult targets due to adversarial training, TESSER + PO outperforms all baselines by significant margins (e.g., + 7% over ATT + PO with ViT-B/16). This highlights that FSGS and SSR lead to perturbations that survive stochastic augmentations while preserving transferability and robustness.

## C.3 ADVERSARIAL ATTACK EFFICIENCY AND CONFIDENCE DYNAMICS

To better assess the quality of adversarial examples beyond final attack success rate (ASR), we evaluate the efficiency and effectiveness of the generated perturbations in terms of iteration-wise model response. Specifically, we compare TESSER and ATT based on:

- **Attack efficiency**: How quickly the model's prediction flips and stabilizes to an adversarial label across iterations.
- **Attack effectiveness**: The final confidence of the model in the adversarial label after optimization completes.

Table 10 summarizes the average iteration at which the target model stabilizes on the adversarial label (i.e., no further label flipping) and the average confidence on the adversarial class after 10 attack steps.

Table 10: Comparison of attack efficiency and effectiveness between TESSER and ATT. We report the average iteration where the model prediction stabilizes on the adversarial label (lower is better) and the final model confidence (%) in the adversarial class (higher is better).

| Method | Stabilization Iteration ($\downarrow$) | Final Confidence (%) ($\uparrow$) |
|---|---|---|
| ATT | 6.8 | 87.93 |
| TESSER (Ours) | **5.1** | **91.37** |

Table 11: TESSER attack success rate (%) with and without module-wise gradient weakening $\omega$ against eight ViT models and the average attack success rate (%) of all black-box models. The best results are highlighted in **bold**.

| Model | Attack | ViT-B/16 | PiT-B | CaiT-S/24 | Visformer-S | DeiT-B | TNT-S | LeViT-256 | ConViT-B | Avg$_{bb}$ |
|---|---|---|---|---|---|---|---|---|---|---|
| ViT-B/16 | w/o $\omega$ | **99.6*** | 40.9 | 71 | 45.7 | 69.3 | 64.8 | 40.2 | 72.5 | 63 |
| | w $\omega$ | **100*** | **61.7** | **94** | **68.3** | **92.5** | **85.6** | **72.2** | **91.4** | **83.2**$\uparrow$ |
| PiT-B | w/o $\omega$ | 58.1 | 99.9* | 69.2 | 74.7 | 69 | 74 | 66.9 | 70.3 | 72.76 |
| | w $\omega$ | **74.9** | **100.0*** | **91.6** | **93.2** | **92.1** | **95** | **92.4** | **91.7** | **91.4**$\uparrow$ |

TESSER reaches a stable adversarial label approximately 1.7 iterations earlier than ATT, confirming its improved gradient alignment and optimization direction. Additionally, the final adversarial confidence achieved by TESSER is consistently higher, indicating stronger and more decisive misclassification. This validates that our semantic gradient modulation not only accelerates convergence but also increases attack effectiveness by pushing perturbations toward model-relevant, transferable features.

# D  ADDITIONAL ABLATION STUDIES

## D.1  QUALITATIVE COMPARISON

To further analyze the effectiveness and interpretability of our proposed method, we present qualitative comparisons between TESSER (FSGS+SSR) and the state-of-the-art ATT (Ming et al., 2024) across a diverse set of samples from ImageNet. Figure 3 shows clean and adversarial images, Grad-CAM heatmaps, and FFT visualizations for perturbations.

**Semantic Alignment.**  In nearly all examples, the adversarial images generated by TESSER show stronger alignment with semantically meaningful regions (e.g., bird bodies, faces, objects of interest) compared to ATT. This is reflected in the Grad-CAM visualizations guided by the adversarial label. Despite being misclassified, the Grad-CAM of TESSER adversarial examples remains spatially focused on relevant visual features, validating the effectiveness of FSGS in preserving semantically informative gradients during attack optimization.

**Spectral Coherence.**  The FFT visualizations reveal that TESSER perturbations exhibit smoother and more coherent frequency profiles, with lower high-frequency energy content than those generated by ATT. This is consistently supported by the computed High-Frequency Energy Ratio (HFER), which is reduced by 6–16% across examples when using FSGS+SSR. Lower HFER confirms that SSR suppresses architecture-specific, high-frequency noise that often undermines transferability.

These additional ablations reinforce our core claim: FSGS guides perturbations toward transferable, semantically meaningful features, while SSR regularizes their spectral profile to avoid overfitting to model-specific noise. Together, these properties lead to adversarial examples that are more interpretable and more effective in black-box transfer scenarios.

## D.2  IMPACT OF MODULE-WISE GRADIENT WEAKENING

We compare ASR with and without $\omega$ (i.e., setting all $\omega = 1$ disables gradient weakening). On ViT-B/16, using $\omega$ improves ASR from 63.0% to 83.2% (Table 11), confirming the effectiveness of selective gradient suppression.

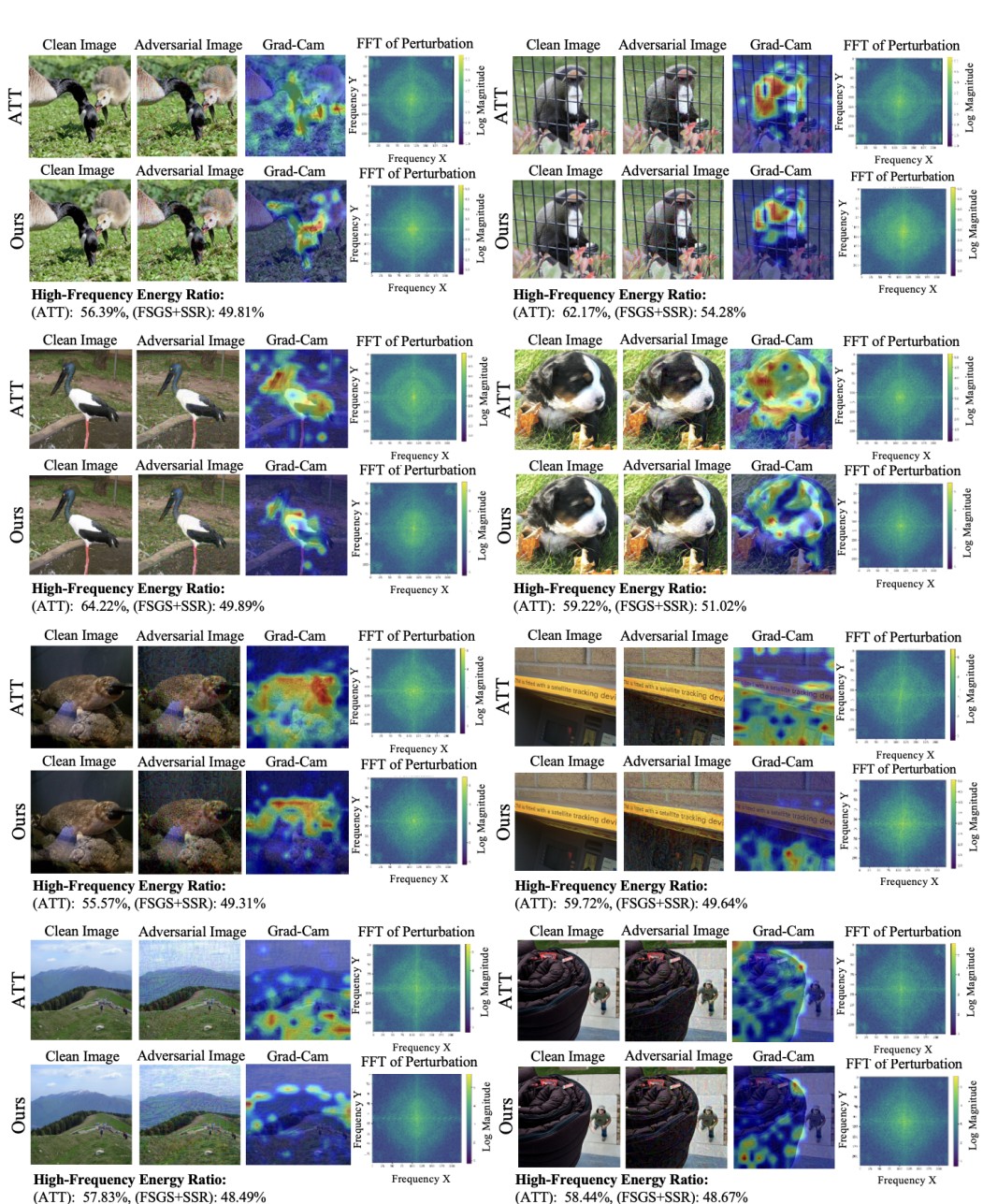

Figure 3: Qualitative comparison between ATT and our TESSER method (FSGS+SSR). Each block shows clean image, adversarial image, Grad-CAM heatmap, and FFT of the perturbation. TESSER yields semantically aligned and spectrally smooth perturbations, with consistently lower high-frequency energy ratios.

Table 12: TESSER attack success rate (%) with and without selective attention truncation $l_{\text{cut}}$ against eight ViT models and the average attack success rate (%) of all black-box models. The best results are highlighted in **bold**.

| Model | Attack | ViT-B/16 | PiT-B | CaiT-S/24 | Visformer-S | DeiT-B | TNT-S | LeViT-256 | ConViT-B | Avg$_{bb}$ |
|---|---|---|---|---|---|---|---|---|---|---|
| ViT-B/16 | w/o $l_{\text{cut}}$ | **100*** | 60.4 | 87.3 | 67.7 | 88.3 | 85.4 | 65.4 | 89.9 | 80.55 |
| | w $l_{\text{cut}}$ | **100*** | **61.7** | **94** | **68.3** | **92.5** | **85.6** | **72.2** | **91.4** | **83.2**↑ |
| PiT-B | w/o $l_{\text{cut}}$ | 73.1 | **99.7*** | 82.1 | 86 | 84 | 86.4 | 83.2 | 84.1 | 84.82 |
| | w $l_{\text{cut}}$ | **74.9** | **100.0*** | **91.6** | **93.2** | **92.1** | **95** | **92.4** | **91.7** | **91.4**↑ |

Table 13: TESSER attack success rate (%) with and without rescaling factor $\lambda$ against eight ViT models and the average attack success rate (%) of all black-box models. The best results are highlighted in **bold**.

| Model | Attack | ViT-B/16 | PiT-B | CaiT-S/24 | Visformer-S | DeiT-B | TNT-S | LeViT-256 | ConViT-B | Avg$_{bb}$ |
|---|---|---|---|---|---|---|---|---|---|---|
| ViT-B/16 | w/o $\lambda$ | **84.1*** | 32.1 | 54.5 | 42.3 | 55.1 | 56 | 43.1 | 56.9 | 53.01 |
| | w $\lambda$ | **100*** | **61.7** | **94** | **68.3** | **92.5** | **85.6** | **72.2** | **91.4** | **83.2**↑ |
| PiT-B | w/o $\lambda$ | 51.2 | **92.9*** | 61.2 | 67.9 | 63.6 | 67.9 | 66.5 | 62.6 | 66.72 |
| | w $\lambda$ | **74.9** | **100.0*** | **91.6** | **93.2** | **92.1** | **95** | **92.4** | **91.7** | **91.4**↑ |

### D.3 Impact of Selective Attention Truncation

On PiT-B, disabling attention truncation (i.e., no $l_{\text{cut}}$) leads to an average 7% drop in ASR (Table 12), validating the importance of focusing on early-layer token gradients.

### D.4 Impact of rescaling factor

Setting $\lambda = 0$ disables adaptive FSGS scaling (only $\gamma_{base}$ is used as a fixed multiplier). On ViT-B/16, enabling $\lambda$ improves ASR by an average of 30% (Table 13), highlighting the value of adaptive gradient modulation in improving attack effectiveness.

In addition, we empirically validate the effectiveness of our scaling strategy by comparing it to a random scaling baseline. As shown in Table 14, our method significantly outperforms random scaling across all target models, achieving consistently higher ASR and demonstrating stronger transferability.

## E Evaluating the Transferability of Different Attack Methods for Targeted Attacks

While our main experiments focus on untargeted attacks, both FSGS and SSR are model-agnostic and loss-independent components applied during backpropagation. Therefore, they are fully compatible with targeted attack formulations—only the loss needs to be adapted. To validate this, we conducted targeted attack experiments using the target label set as (true label + 1). As shown in Table 15, our method (TESSER) achieves a significantly higher targeted ASR of 43.08%, outperforming PGD (16.06%), MIM (22.01%), and ATT (33.33%), demonstrating the effectiveness and transferability of our approach in targeted settings as well. The table will be included in the revised version.

Table 14: TESSER Attack Success Rate (%) with our Scaling strategy vs. Random Scaling. *Bold* = better of the two scalings for the same surrogate–target pair. * denotes white-box (surrogate equals target).

| Surrogate | Scaling | ViT-B/16 | PiT-B | CaiT-S/24 | Visformer-S | DeiT-B | TNT-S | LeViT-256 | ConViT-B | Avg |
|---|---|---|---|---|---|---|---|---|---|---|
| ViT-B/16 | Random | **86.4*** | 30.6 | 52.3 | 37.8 | 53.0 | 55.4 | 37.8 | 55.7 | 51.12 |
| | ours | **100*** | **61.7** | **94.0** | **68.3** | **92.5** | **85.6** | **72.2** | **91.4** | **83.2** ↑ |
| PiT-B | Random | 28.4 | **100*** | 34.6 | 44.8 | 33.9 | 44.1 | 38.3 | 38.2 | 45.28 |
| | ours | **74.9** | **100.0*** | **91.6** | **93.2** | **92.1** | **95.0** | **92.4** | **91.7** | **91.4** ↑ |

Table 15: The attack success rate (%) of various transfer-based targeted attacks against eight ViT models and the average attack success rate (%) of all black-box models. The best results are highlighted in **bold**.

| Model | Attack | ViT-B/16 | PiT-B | CaiT-S/24 | Visformer-S | DeiT-B | TNT-S | LeViT-256 | ConViT-B | Avg$_{bb}$ |
|-------|--------|----------|-------|-----------|-------------|--------|-------|-----------|----------|-----------|
| ViT-B/16 | PGD | **96.1*** | 2.1 | 6.8 | 2.7 | 5.6 | 6 | 1.7 | 7.5 | 16.06 |
|  | MIM | **99.4*** | 6.6 | 16.1 | 6.3 | 13.9 | 11.8 | 4.4 | 17.6 | 22.01 |
|  | ATT | **99.5*** | 8.7 | 35.6 | 9.6 | 33.6 | 30.9 | 6.7 | 42.1 | 33.33 |
|  | Ours | **99.6*** | **17.9** | **47.3** | **21** | **48.6** | **42.1** | **16.1** | **52.1** | **43.08**↑ |
| PiT-B | PGD | 0.8 | **96.6*** | 1.1 | 2.3 | 0.9 | 2.1 | 1.3 | 0.7 | 13.22 |
|  | MIM | 5.3 | **99.9*** | 5.1 | 8 | 4.9 | 6.5 | 4 | 5.5 | 17.4 |
|  | ATT | 12.1 | **100*** | 14.6 | 20.2 | 13.2 | 18.8 | 12.2 | 16.8 | 25.98 |
|  | Ours | **20.4** | **100.0*** | **26** | **33** | **28.4** | **30.2** | **26.3** | **27.2** | **47.86**↑ |

Table 16: The attack success rate (%) of Autoattack (AA) vs. TESSER against eight ViT models and the average attack success rate (%) of all black-box models. The best results are highlighted in **bold**.

| Model | Attack | ViT-B/16 | PiT-B | CaiT-S/24 | Visformer-S | DeiT-B | TNT-S | LeViT-256 | ConViT-B | Avg$_{bb}$ |
|-------|--------|----------|-------|-----------|-------------|--------|-------|-----------|----------|-----------|
| ViT-B/16 | AA | **99.9*** | 10.5 | 42.9 | 13.2 | 33.6 | 38.4 | 17 | 40.4 | 36.98 |
|  | Ours | **100*** | **61.7** | **94** | **68.3** | **92.5** | **85.6** | **72.2** | **91.4** | **83.2**↑ |
| PiT-B | AA | 10 | **98.5*** | 13.5 | 24 | 11.3 | 21.2 | 22.4 | 13.9 | 26.85 |
|  | Ours | **74.9** | **100.0*** | **91.6** | **93.2** | **92.1** | **95** | **92.4** | **91.7** | **91.4**↑ |

## F  EVALUATING TESSER PERFORMANCE VS. AUTOATTACK

While AutoAttack (AA) is a strong white-box evaluation benchmark, it is not optimized for transfer-based black-box settings. To enable a fair comparison, we evaluate both TESSER and AutoAttack under the same transfer setup with a fixed perturbation budget of epsilon = 16/255. As reported in Table 16, TESSER achieves over 50% higher average ASR compared to AutoAttack across multiple target models. For example, when attacking PiT-B from a ViT-B/16 surrogate, TESSER achieves an ASR of 61.7%, compared to only 10.5% for AutoAttack. This gap is expected, as TESSER is explicitly designed to optimize black-box transferability, whereas AutoAttack is tailored for white-box robustness evaluation.

## G  EVALUATION TESSER TRANSFERABILITY TO VISUAL STATE SPACE MODELS

To further increase the architectural dissimilarity, we evaluate TESSER transferability to Vision Mamba (Zhu et al., 2024), a state-space-based architecture with bidirectional SSMs and position, aware embeddings, representing a class of models distinct from transformers. As shown in Table 17, TESSER consistently achieves the highest ASR 87.1% and 76.7% on both Vim-Tiny and Vim-Small, respectively compared to 80.9% and 69% for sota ATT attack, demonstrating robust transfer even under significant architectural and representational divergence.

Table 17: Comparative experiments of different attack methods on VIM. "clean" indicates that clean images are classified and all results indicate the percentage of classification errors (*i.e.*, ASR).

| Model | Attack | VIM-tiny | VIM-small |
|-------|--------|----------|-----------|
| ViT-B/16 | clean | 3.1 | 0.9 |
|  | MIM | 45.6 | 42 |
|  | ATT | 80.9 | 69 |
|  | TESSER | **87.1**↑ | **76.7**↑ |
| PiT-B | MIM | 32.3 | 34.9 |
|  | ATT | 53.4 | 55.1 |
|  | TESSER | **77.4**↑ | **80.7**↑ |

## H  SSR EFFECTIVENESS IN THE FREQUENCY DOMAIN

### H.1  CONTROLLED BAND-PASS STRESS TEST (SPECTRAL ANALYSIS OF PERTURBATIONS)

To quantitatively evaluate the spectral behavior introduced by the proposed **Spectral Smoothness Regularization (SSR)** and to compare it with existing baselines (**ATT** and **TGR**), we design a controlled frequency-domain stress test that isolates the contribution of different spectral bands in adversarial perturbations.

**Experimental Setup.**    For each attack method, we compute perturbations $\delta = x_{\text{adv}} - x_{\text{clean}}$ and apply frequency-domain filtering prior to reconstruction. Specifically, we generate ideal circular *low-pass (LP)* and *high-pass (HP)* masks in the Fourier domain with normalized cutoff radii $r \in \{0.05, 0.1, 0.2, 0.3, 0.5\} \times r_{\text{max}}$. Each perturbation is filtered as

$$\tilde{\delta} = \mathcal{F}^{-1}(\mathcal{M} \odot \mathcal{F}(\delta)),$$

where $\mathcal{M}$ is the LP or HP mask and $\mathcal{F}$ denotes the 2D Fourier Transform. The filtered perturbation is then projected back into the valid $L_\infty$-bounded region, producing new adversarial examples $x' = \text{clip}(x_{\text{clean}} + \tilde{\delta})$. We then re-evaluate the *Attack Success Rate (ASR)* of each filtered adversarial image on the target model. This controlled procedure directly measures how strongly each attack relies on specific frequency bands.

The measured ASR (%) across cutoff frequencies for both HP and LP masks is summarized in Table 18.

| **Method** | **HP (High-Pass Filter)** | | | | | **LP (Low-Pass Filter)** | | | | |
|---|---|---|---|---|---|---|---|---|---|---|
| | 0.05 | 0.10 | 0.20 | 0.30 | 0.50 | 0.05 | 0.10 | 0.20 | 0.30 | 0.50 |
| ATT | 100.0 | 98.8 | 64.9 | 15.0 | 0.5 | 0.0 | 2.0 | 19.4 | 75.5 | 99.7 |
| TESSER (SSR) | 99.2 | 97.5 | 88.7 | 59.7 | 3.5 | 4.2 | 23.2 | 77.5 | 95.4 | 99.6 |
| TGR | 99.8 | 99.6 | 96.7 | 83.2 | 15.9 | 3.9 | 17.8 | 60.6 | 88.4 | 99.5 |

Table 18: ASR (%) under controlled frequency-domain filtering. HP columns show the effect of retaining only high-frequency components, while LP columns show the effect of retaining only low-frequency components. Cutoff values correspond to normalized frequency radii ($r/r_{\text{max}}$).

A high ASR under HP filtering indicates that the attack relies on fine-grained, noise-like high-frequency content, while a high ASR under LP filtering suggests that the attack's discriminative power comes from smoother, structured low-frequency variations. Several key observations can be made:

- **ATT:** exhibits a strong high-frequency bias. Its ASR collapses rapidly under low-pass filtering (from 99% to 0%), indicating that its perturbations primarily occupy the high-frequency spectrum.

- **TESSER (with SSR):** maintains significantly higher ASR under LP filtering (77.5% at $r = 0.2$ vs. 19.4% for ATT), confirming that SSR distributes energy across lower and mid-frequency bands and enforces *spectral smoothness*. This evidences that TESSER generates more structured, semantically aligned perturbations rather than high-frequency noise.

- **TGR:** shows intermediate behavior, partially resilient to LP filtering but still decaying faster than TESSER, highlighting that TESSER's SSR achieves stronger low-frequency regularization.

This experiment demonstrates that **Spectral Smoothness Regularization** effectively redistributes perturbation energy from high-frequency, noise-like components to smoother low-frequency structures. Such spectral redistribution validates the intended effect of SSR and correlates with the observed improvements in transferability and robustness across models and quantization levels. Overall, SSR produces *spectrally smoother, semantically coherent perturbations*, whereas ATT remains heavily reliant on fragile high-frequency artifacts.

## H.2 Verifying the Effectiveness of SSR in the Frequency Domain.

We thank the reviewer for encouraging a more controlled spectral analysis. To quantify the frequency redistribution introduced by **Spectral Smoothness Regularization (SSR)**, we computed the relative energy of perturbations from **TESSER**, **ATT**, and **TGR** after applying ideal low-pass (LPF) and high-pass (HPF) filters with a cutoff of $0.25\times$Nyquist. The measured spectral energy distribution is reported below.

| Method | LPF Energy (%) ↓ | HPF Energy (%) ↓ |
|---|---|---|
| ATT | 47.6 | 52.4 |
| TGR | 40.0 | 60.0 |
| TESSER (w/ SSR) | **56.3** | **43.7** |

Table 19: Spectral energy distribution of perturbations under LPF/HPF decomposition (cutoff = $0.25\times$Nyquist).

Compared to ATT and TGR, **TESSER shows a $+8\%$–$+16\%$ increase in low-frequency energy and a $-9\%$–$-16\%$ reduction in high-frequency energy**, confirming that SSR effectively shifts perturbation power toward smoother and more transferable frequency bands. When SSR is disabled, TESSER's cross-architecture ASR on CNNs drops from **74.4%** to **70.3%**, demonstrating that this spectral adjustment materially enhances transferability.

# I Additional Empirical Validation of Semantic and Structural Alignment

## I.1 Empirical Validation of the Semantic Meaning of Token Norms

To validate the core assumption underlying FSGS, we performed a layer-wise analysis measuring the Spearman correlation between token feature norms and token gradient norms across all ViT-B/16 blocks. As shown in Table 20, early layers exhibit weak or even negative correlation, confirming that high-norm shallow tokens do not encode semantic content (consistent with the reviewer's dual-task critique). In contrast, correlation increases steadily through mid-level layers and peaks in deep layers (up to 0.49), indicating that high-norm tokens reliably capture semantic importance precisely where the network forms object-level representations.

This provides direct empirical evidence supporting our design choice: *FSGS does not operate on shallow layers, but applies semantic-aware scaling only in mid-to-deep blocks, where token magnitude meaningfully reflects semantic relevance.*

Table 20: Spearman correlation between token feature norms and token gradient norms across ViT-B/16 layers.

| Layer | 0 | 1 | 2 | 3 | 4 | 5 | 6 | 7 | 8 | 9 | 10 | 11 |
|---|---|---|---|---|---|---|---|---|---|---|---|---|
| **Spearman** $\rho$ | -0.03 | 0.03 | 0.05 | 0.10 | 0.25 | 0.21 | 0.29 | 0.37 | 0.36 | **0.49** | 0.34 | 0.00 |

## I.2 Attention Rollout Analysis (ViT-Specific Semantic Alignment)

To directly address the reviewer's concern regarding the suitability of Grad-CAM for ViTs, we evaluate semantic alignment using *Attention Rollout* (Abnar & Zuidema, 2020), a ViT-native interpretability method. For each clean image and its adversarial counterpart, we compute the CLS→patch rollout map and measure (i) Spearman correlation and (ii) intersection-over-union (IoU) between the two saliency maps.

Rollout maps are structurally stable due to their cumulative nature across layers and heads, which explains the overall high correlations across all methods. Despite this stability, meaningful relative differences arise across attacks. As shown in Table 21, TESSER achieves the strongest semantic alignment (Spearman = 0.9894, IoU = 0.8258), outperforming ATT (0.9581 / 0.8045) and substantially exceeding TGR (0.9322 / 0.6596). These findings demonstrate that *TESSER preserves global*

*semantic attention flow more faithfully than prior methods*, complementing our Grad-CAM results and providing ViT-specific evidence for the semantic coherence of TESSER's perturbations.

Table 21: Attention Rollout–based semantic alignment between clean and adversarial images. Higher is better.

| Method | Rollout Spearman | Rollout IoU |
|--------|------------------|-------------|
| ATT | 0.9581 | 0.8045 |
| TGR | 0.9322 | 0.6596 |
| **TESSER** | **0.9894** | **0.8258** |