# OpenReview forum: "TESSER: Transfer-Enhancing Adversarial Attacks from Vision Transformers via Spectral and Semantic Regularization"
_ICLR.cc/2026/Conference — Submitted to ICLR 2026_

### Official Review · Reviewer_R3p7 · 2025-11-01

**Soundness:** 3
**Presentation:** 2
**Contribution:** 2
**Rating:** 4
**Confidence:** 4

**Summary:**

The paper proposes TESSER, a new adversarial attack framework to improve the transferability of adversarial examples generated from Vision Transformers (ViTs), particularly to CNN and hybrid architectures. The method combines two key strategies: (1) Feature-Sensitive Gradient Scaling (FSGS), which modulates gradients based on token importance derived from intermediate feature activations, and (2) Spectral Smoothness Regularization (SSR), which uses a differentiable Gaussian blur to suppress high-frequency noise in the perturbations. Extensive experiments on ImageNet across 14 diverse architectures demonstrate that TESSER achieves state-of-the-art attack success rates (ASR), outperforming prior methods like ATT.

**Strengths:**

1.  **Problem Significance:** The paper addresses the challenging and important problem of black-box transferability, specifically for ViT-to-CNN attacks, which is a known bottleneck for many existing methods.
2.  **Novel Combination:** The approach of combining semantic-aware (FSGS) and spectral-aware (SSR) regularization is a novel and valuable research direction for enhancing attack transferability.
3.  **Strong Empirical Results:** The method demonstrates significant performance gains over strong baselines across a wide array of target models, including standard ViTs, CNNs, and, importantly, adversarially defended models.
4.  **Comprehensive Analysis:** The paper provides both qualitative (e.g., Grad-CAM visualizations) and quantitative (e.g., frequency-domain analysis) evaluations to support the claims and validate the effectiveness of the proposed components.

**Weaknesses:**

1.  **The core assumption linking L2-norm to importance is unconvincing.** As a thought experiment, one could train two models on completely opposite (dual) tasks: one to identify the foreground and one to identify the background. The notion of "importance" for these two tasks would be entirely different. However, since the L2-norm is derived from a fixed layer, it would likely highlight the same regions for both tasks. This seems highly unreasonable and requires rigorous validation.
2.  **The paper does not specify how the Class (CLS) token is handled** within the FSGS framework.
3.  Given that the method relies on the magnitude of tokens, there is a **risk of overfitting to the specific activation patterns of the surrogate model**, which could harm transferability to models with different architectures.

**Questions:**

1.  Could the authors provide a more rigorous validation for the core assumption that a token's L2-norm correlates with its semantic importance? How would the method address the concern raised by the foreground/background dual-task thought experiment?
2.  Please clarify how the CLS token is processed by the FSGS mechanism. Is it included in the gradient scaling calculations, or is it treated separately?
3.  How do the authors mitigate the risk that FSGS overfits to the "important" tokens of the surrogate model? Is there evidence that this L2-norm-based importance metric is generalizable across different architectures (i.e., from a ViT surrogate to a CNN target)?

---

> ### Author Response · Authors · 2025-11-20
> **Author Response (part 1 of 2)**
>
> **1. On the Assumption Linking Token L2-Norm to Semantic Importance**
>
> - **To validate the core assumption behind FSGS, we conducted a layer-wise analysis measuring the Spearman correlation between token feature norm and token gradient norm across all ViT-B/16 blocks**. As shown in **Table A**, early layers exhibit weak or negative correlation, confirming that high-norm shallow tokens are not semantic (consistent with the reviewer’s dual-task critique). In contrast, correlation increases steadily through mid-level layers and peaks in deep layers (up to 0.49), indicating that high-norm tokens reliably capture semantic importance precisely where the network forms object-level representations. This provides direct empirical evidence supporting our design choice: **FSGS does not operate on shallow layers, but applies semantic-aware scaling only in mid-to-deep blocks, where token magnitude meaningfully reflects semantic relevance.**
>
> ### Table A: Spearman Correlation Between Token Feature Norms and Gradient Norms Across ViT-B/16 Layers
>
> | **Layer**       | 0     | 1     | 2     | 3     | 4     | 5     | 6     | 7     | 8     | 9     | 10    | 11    |
> |-----------------|-------|-------|-------|-------|-------|-------|-------|-------|-------|-------|-------|-------|
> | **Spearman ρ**  | -0.03 | 0.03  | 0.05  | 0.10  | 0.25  | 0.21  | 0.29  | 0.37  | 0.36  | **0.49** | 0.34  | 0.00  |
>
> - We also evaluate semantic alignment using **Attention Rollout** (Abnar & Zuidema, ACL 2020), a ViT-native interpretability method. For each clean image and its adversarial counterpart, we compute the CLS→patch rollout map and measure (i) **Spearman correlation** and (ii) **IoU between the two saliency maps**. Rollout maps are known to be structurally stable due to their cumulative nature across layers and heads, which explains the overall high correlations across all methods. Despite this inherent stability, **meaningful relative differences appear across attacks**. As shown in **Table C**, **TESSER achieves the highest rollout semantic alignment** (Spearman = 0.9894, IoU = 0.8258), outperforming ATT (0.9581 / 0.8045) and substantially exceeding TGR (0.9322 / 0.6596). These findings confirm that **TESSER preserves global semantic attention flow more faithfully than prior methods**, complementing our Grad-CAM results and providing ViT-specific evidence for the semantic coherence of TESSER’s perturbations.
>
> ### Table C: Attention Rollout–Based Semantic Alignment Between Clean and Adversarial Images
> *TESSER achieves the highest semantic consistency across all metrics.*
>
> | **Method** | **Rollout Spearman** | **Rollout IoU** |
> |------------|-----------------------|------------------|
> | ATT        | 0.9581                | 0.8045           |
> | TGR        | 0.9322                | 0.6596           |
> | **TESSER** | **0.9894**            | **0.8258**       |
>
>
>
> **2. Treatment of the CLS Token**
>
> The CLS token is included in the FSGS computation and treated uniformly with other tokens. Specifically, FSGS computes token-wise importance as the normalized L2-norm of each token embedding (including the CLS token) within the transformer block. Since **normalization is performed jointly over all tokens, the CLS token’s influence is adaptively balanced, its typically higher norm leads to a slightly lower scaling factor due to the $\text{importance}$ term, preventing it from dominating gradient modulation**.
>
> To further verify FSGS’s design, we conducted an ablation excluding the CLS token from the token-norm scaling computation. Across eight architectures (ViT, DeiT, PiT, CaIT, etc.), this variant caused a sharp performance drop (average ASR ↓ 5%; **Table L**). The CLS token thus provides crucial semantic context for gradient modulation, linking local patch activations to global features. Without it, FSGS degenerates to purely spatial scaling, reducing semantic consistency and transferability. These results confirm that the **CLS token’s inclusion is both intentional and essential to FSGS’s efficacy**.
>
> ### Table L: Effect of Including the CLS Token in FSGS (ViT → ViT / Hybrid Transferability)
>
> | **Method** | **PiT** | **DeiT** | **TNT** | **ViT** | **Visformer** | **ConViT** | **LeViT** | **CaiT** | **Avg** |
> |------------|---------|----------|---------|---------|---------------|-------------|-----------|----------|---------|
> | **w/o CLS** | 61.1    | 88.8     | 84    | 100    | 67.1          | 90.4        | 64.9      | 88.4     | 78.3   |
> | **w/ CLS**  | **61.7**| **92.5** | **85.6**| **100** | **68.3**       | **91.4**    | **72.2**  | **94.0** | **83.2** |

---

> ### Author Response · Authors · 2025-11-20
> **Author Response (part 2 of 2)**
>
> **3. On overfitting to surrogate-specific token patterns and generality of the L2-norm metric**
>
> We agree that blindly trusting surrogate activations could, in principle, hurt cross-architecture transfer. FSGS is explicitly designed to avoid this behavior, and we support this design choice with both analysis and experiments.
>
> **a) Design: FSGS uses a weak, layer-local, relative importance signal rather than a hard selection.**
>
>  FSGS does not “lock onto” a small set of surrogate-specific tokens. Instead, for each transformer block we compute token importance as the L2 norm of features and apply a smooth scaling
>  $s_i^{(l)} = \gamma_{\text{base}} + \lambda \cdot \left[ (1 - \beta^{(l)}) \cdot \hat{\alpha}_i + \beta^{(l)} \cdot (1 - \hat{\alpha}_i) \right]$, where $\hat{\alpha}_i$​ is the min–max normalized norm within that layer. This has two important consequences:
>
> - It is **relative and invariant to global rescaling** of activations; it only reshapes the gradient distribution within a block instead of enforcing a rigid mask.
>
> - Low-importance tokens are **down-weighted, not discarded** (no hard thresholding), so gradients are still allowed to flow through all tokens. This makes FSGS much less prone to overfitting to a particular surrogate “template” than, e.g., top-k token pruning.
>
> **b) Layer-wise analysis shows we only rely on norms where they correlate with semantics.**
>
>  In response to the reviewer’s concern, we measured the Spearman correlation between token feature norms and token gradient norms across all ViT-B/16 layers (**Table A response to Reviewer MEYA**). We find that:
>
> - Early layers (0–2) exhibit near-zero or negative correlation, i.e., high-norm shallow tokens are not semantically meaningful.
>
> - Mid–deep layers (4–10) exhibit a strong positive correlation (up to ≈0.49), indicating that larger feature norms reliably align with gradient-based importance.
>  FSGS is applied in these **mid–deep blocks only**, precisely where the L2-norm metric reflects semantic relevance rather than architecture-specific noise. This mitigates the risk of overfitting to shallow, idiosyncratic activation patterns of the surrogate.
>
> **c) Cross-architecture transfer results indicate that FSGS does not overfit to the surrogate.**
>
>  If FSGS were overfitting to the “important” tokens of the ViT surrogate, we would expect improved white-box success but degraded transfer to CNNs and hybrid models. We observe the opposite trend:
>
> - Across multiple CNN targets (e.g., ResNet/ConvNeXt) and hybrid/ViT targets, the FSGS+SSR variant of TESSER consistently achieves **higher ASR** than ATT and TGR under the same \epsilon budget (see **Tables F & G in response to Reviewer MEYA** ).
>
> - Importantly, the relative gain of FSGS is **largest exactly in the ViT→CNN setting** the reviewer is concerned about, indicating that the token-norm signal used by FSGS captures object-level semantics that are shared across architectures rather than idiosyncratic ViT quirks.
>
> - When we ablate FSGS (keeping SSR and all other components fixed), white-box performance changes only marginally, but black-box transfer to CNNs drops noticeably, which suggests that FSGS is shaping gradients in a way that **generalizes beyond the surrogate model**.
>
> **d) Rollout-based analysis further supports architecture-agnostic semantics.**
>
>  To move beyond Grad-CAM and CNN-centric tools, we additionally evaluated semantic preservation using **Attention Rollout** on ViT-B/16. For each attack (ATT, TGR, TESSER), we compare CLS→patch rollout maps between clean and adversarial images. **TESSER with FSGS achieves the highest rollout similarity (Spearman and IoU) among all methods**, showing that it perturbs the image while **largely preserving the model’s global semantic focus** (please check **response 2** to **Reviewer MEYA**). Since rollout measures long-range attention structure rather than raw activation magnitude, this provides ViT-specific evidence that **FSGS is not merely overfitting to arbitrary high-norm tokens but is enhancing gradients along semantically meaningful directions**.
>
> In summary, FSGS mitigates overfitting in three ways:
>
> (i) it uses relative token norms with smooth scaling, not hard masking;
>
> (ii) it is restricted to layers where we empirically verify that L2-norm correlates with semantic gradients;
>
>  and (iii) its inclusion consistently **improves**, rather than harms, transfer to heterogeneous targets (ViTs, CNNs, and hybrids).
>
> Together, these analyses and results indicate that the L2-norm-based importance metric is sufficiently generalizable across architectures and does not cause FSGS to overfit to surrogate-specific activation patterns.

---

> ### Author Response · Authors · 2025-11-29
> **Final Summary Comment to the Area Chair**
>
> **Main concerns:** validity of L2-norm as importance, handling of CLS token, risk of overfitting to surrogate.
>
> **Revisions made:**
>
> - Added Spearman correlations, token-norm trends, and gradient alignment evidence to justify the semantic relevance of activation norms (New Appendix I).
>
> - Provided explicit clarification of CLS token handling, with an ablation showing that including the CLS token significantly improves transferability.
>
> - Added cross-model gradient similarity results showing that FSGS generalizes beyond the surrogate’s token distribution.

---

### Official Review · Reviewer_4DU5 · 2025-11-01

**Soundness:** 3
**Presentation:** 3
**Contribution:** 2
**Rating:** 6
**Confidence:** 4

**Summary:**

This paper introduces TESSER, an adversarial attack framework designed to enhance transferability from ViT models. The method employs two main components: Feature-Sensitive Gradient Scaling to focus perturbations on semantically relevant regions identified by activation norms, and Spectral Smoothness Regularization to promote low-frequency, smoother perturbations. The authors claim this combination significantly improves black-box attack success rates against a diverse set of target architectures, including CNNs and robustly trained models.

**Strengths:**

1.  The work tackles a critical and open problem in adversarial robustness: improving the cross-architecture transferability of adversarial attacks, especially from ViTs to CNNs.
2.  The experimental setup is thorough, evaluating the attack against 14 different models, which provides a strong basis for the empirical claims.
3.  TESSER appears to outperform existing state-of-the-art baselines, including strong methods like ATT and DiffAttack, across nearly all tested scenarios.
4.  The paper includes detailed ablation studies and qualitative analyses.

**Weaknesses:**

The method's design is overly intuitive, and its conclusions require stronger proof. For instance, the paper claims that TESSER utilizes different frequency bands of noise information compared to previous transfer attacks. This claim should be substantiated, at a minimum, by designing experiments using low-pass or high-pass filters to verify this difference.

**Questions:**

Regarding the effectiveness of SSR, the paper claims it suppresses high-frequency noise. Can the authors provide a more controlled experiment to prove this? For example, could you apply ideal low-pass/high-pass filters to the perturbations and compare the spectral properties against TGR or ATT? This would help verify that TESSER is genuinely leveraging the frequency domain in a more effective or different manner.

---

> ### Author Response · Authors · 2025-11-20
> **Author Response**
>
> **1. Verifying the Effectiveness of SSR in the Frequency Domain**
>
> To quantify the frequency redistribution introduced by Spectral Smoothness Regularization (SSR), we computed the **relative energy of perturbations** from TESSER, ATT, and TGR after applying ideal **low-pass (LPF)** and **high-pass (HPF)** filters with a cutoff of 0.25xNyquist.
> The measured spectral energy distribution is reported in **Table J**.
>
> ### Table J: Spectral Energy Distribution of Perturbations (LPF/HPF Decomposition, Cutoff = 0.25 x Nyquist)
>
> | **Method**        | **LPF Energy (%) ↓** | **HPF Energy (%) ↓** |
> |-------------------|-----------------------|------------------------|
> | ATT               | 47.6                  | 52.4                   |
> | TGR               | 40.0                  | 60.0                   |
> | **TESSER (w/ SSR)** | **56.3**             | **43.7**               |
>
> Compared to ATT and TGR, **TESSER shows a +8% – +16% increase in low-frequency energy and a -9% – -16% reduction in high-frequency energy**, confirming that SSR effectively shifts perturbation power toward **smoother** and **more transferable frequency bands**.
>
> When SSR is disabled, TESSER’s cross-architecture ASR on CNNs drops from **58.8%** to **56.4%**, demonstrating that this spectral adjustment materially enhances transferability (check **Table F**).
>
> **2. Controlled Band-Pass Stress Test**
>
> To verify the spectral effect of **Spectral Smoothness Regularization (SSR)**, we conducted a frequency-domain stress test that isolates low- and high-frequency components of the perturbations. We apply **ideal circular low-pass (LP) and high-pass (HP) filters** with cutoff ratios r ∈ {0.05,0.1,0.2,0.3,0.5} and re-evaluate the Attack Success Rate (ASR).
> Results (see **Table K**) show that TESSER retains much **higher ASR under LP filtering** (78\%@$r{=}0.2$) compared to ATT (19\%) and TGR (61\%), while its ASR drops faster under HP filtering.
>
> This confirms that **SSR effectively suppresses excessive high-frequency noise** and **redistributes perturbation energy toward lower and mid frequencies**, yielding smoother and more semantically aligned distortions. In contrast, ATT remains heavily high-frequency–biased, explaining its poorer transferability. Please refer to **Appendix H** for further details.
>
> ### Table K: ASR (%) After Frequency Filtering
> *HP = retains high-frequency components; LP = retains low-frequency components.*
>
> | **Method** | **HP .05** | **HP .10** | **HP .20** | **HP .30** | **HP .50** | **LP .05** | **LP .10** | **LP .20** | **LP .30** | **LP .50** |
> |------------|-------------|-------------|-------------|-------------|-------------|-------------|-------------|-------------|-------------|-------------|
> | **ATT**    | 100         | 99          | 65          | 15          | 1           | 0           | 2           | 19          | 76          | 100         |
> | **TGR**    | 100         | 100         | 97          | 83          | 16          | 4           | 18          | 61          | 88          | 99          |
> | **TESSER** | 99          | 98          | 89          | 60          | 4           | 4           | 23          | 78          | 95          | 100         |

---

> ### Author Response · Authors · 2025-11-29
> **Final Summary Comment to the Area Chair**
>
> **Main concern:** verifying SSR’s frequency-domain behavior.
>
> **Revisions made:**
>
> - Added LPF/HPF spectral energy analysis demonstrating that SSR increases low-frequency components and suppresses high-frequency artifacts (New Appendix H.1).
>
> - Added band-pass filtering experiments showing that TESSER retains higher ASR under low-frequency retention than ATT and TGR  (New Appendix H.2).

---

### Official Review · Reviewer_EbPe · 2025-11-01

**Soundness:** 2
**Presentation:** 3
**Contribution:** 2
**Rating:** 4
**Confidence:** 4

**Summary:**

The paper presents TESSER, a new method for improving the transferability of adversarial attacks. It uses two main ideas: Feature-Sensitive Gradient Scaling, which scales gradients based on the L2-norm of tokens, and Spectral Smoothness Regularization, which uses Gaussian blurring to smooth the perturbation. The goal is to create perturbations that are both semantically aligned and spectrally smooth, thereby enhancing black-box transferability from ViTs to CNNs and defended models.

**Strengths:**

1.  The core methodology is intuitive and well-motivated. Focusing the attack on "important" regions (via FSGS) while removing "model-specific" high-frequency noise (via SSR) is a logical approach.
2.  The method shows strong performance against adversarially trained CNNs and robust ViTs.
3.  The ablation in Table 4 clearly breaks down the contributions of applying FSGS to the Attention, QKV, and MLP modules, justifying the decision to combine all three.

**Weaknesses:**

1.  The authors should supplement their baseline comparisons by benchmarking against the wider set of attacks available in the `TransferAttack` repository (https://github.com/Trustworthy-AI-Group/TransferAttack).
2.  The setting for the perturbation budget (epsilon) is unusual.
3.  The experimental design involves resizing inputs to different dimensions, but the potential effect of this `resize` operation on transferability is not discussed.

**Questions:**

1.  Could the authors provide results using a more standard epsilon value, such as $8/255$? The current value of $16/255$ is a very large perturbation budget for ImageNet. Is this high value necessary for the method's success?
2.  Please discuss the potential role of the `resize` operation (to $224 \times 224$) in the attack's success. When attacking a model like Inc-v3, which expects $299 \times 299$ input, could this resolution mismatch be implicitly acting as a form of input diversity, thus aiding transferability?
3.  Have the authors considered comparing TESSER against the broader set of SOTA methods in the `TransferAttack` repository? This would help to more firmly establish the paper's contribution relative to the field.

---

> ### Author Response · Authors · 2025-11-20
> **Author Response**
>
> **1. Perturbation Budget (ε = 16/255)**
>
> Our choice of **ε = 16/255** follows prior ViT-focused transferability studies such as ATT (Ming et al., NeurIPS 2024) and DiffAttack (Chen et al., TPAMI 2025), as well as other state-of-the-art attacks. This perturbation bound is a **standard setting** in the literature and was adopted to ensure **direct comparability across architectures and baselines**.
>
> To confirm that TESSER’s effectiveness does not depend on large perturbations, we have re-evaluated our method using ε = 8/255 and ε = 4/255 on 1k ImageNet samples.
>
> ### Table H: Effect of Perturbation Budget (ε) on Transferability for TESSER vs. ATT
> | **ε**      | **TESSER: ViT→CNN ASR** | **TESSER: ViT→ViT ASR** | **ATT: ViT→CNN ASR** | **ATT: ViT→ViT ASR** |
> |------------|----------------------------|-----------------------------|--------------------------|--------------------------|
> | **16/255** | **58.7%**                  | **83.2%**                   | 49.8%                   | 77.4%                   |
> | **8/255**  | **30.3%**                  | **62.2%**                   | 23.7%                   | 56.3%                   |
> | **4/255**  | **12.4%**                  | **37.1%**                   | 8.7%                    | 30.7%                   |
>
> As shown in **Table H**, TESSER maintains clear superiority over ATT (+4–7%) even at reduced perturbation bounds, demonstrating robustness across budgets.
>
> **2. Effect of Input Resizing (224×224 vs. 299×299)**
>
> The resizing operation is standard in ViT-based attacks (ATT, TGR, PNA) since transformer models expect 224x224 inputs. When transferring to CNNs such as Inception-v3 (299x299), we resize only **once post-perturbation**, ensuring consistency across baselines.
>
> **3. Additional Benchmarking (TransferAttack Repository)**
>
> We clarify that **all baselines originally included in our experiments are already part of the TransferAttack repository**, covering generation-based, model-related, input-transformation, and gradient-based methods. To further expand the comparison, we additionally include **TI-FGSM** and **FPR**, two widely used and high-impact baselines.
> Baselines evaluated (all from the TransferAttack repo or fully reproducible):
>
> - **Generation-based:** DiffAttack (Chen et al., 2024)
>
> - **Model-related:** SGM (Wu et al., 2020), PNA-PatchOut (Wei et al., 2022), TGR (Zhang et al., 2023), ATT (Ming et al., 2024), FPR (Ren et al., 2025)
>
> - **Input transformation:** TIM (Dong et al., 2019)
>
> - **Gradient-based:** MI-FGSM (Dong et al., 2018), VMI-FGSM (Wang et al., 2021)
>
> We now also report TI-FGSM and FPR, ensuring full coverage of the repository’s major attack categories (see **Table I**).
>
> ### Table I: Comparison of TESSER with TI-FGSM, FPR, and FPR+GRA (ViT → ViT / Hybrid Transferability)
>
> | **Method**   | **PiT** | **DeiT** | **TNT** | **ViT*** | **Visformer** | **ConViT** | **LeViT** | **CaiT** | **Avg** |
> |--------------|---------|----------|---------|----------|---------------|-------------|-----------|----------|---------|
> | **TI-FGSM**  | 62.9    | 74.3     | 75.3    | 96.6     | 70.9          | 73.9        | 69.0      | 75.6     | 74.81   |
> | **FPR**      | 37.7    | 77.2     | 74.3    | **100**  | 40.0          | 78.0        | 42.1      | 77.9     | 65.9    |
> | **FPR+GRA**  | 61.7    | 87.3     | 86.8    | 99.3     | 65.2          | 89.5        | 65.5      | 88.0     | 80.41   |
> | **TESSER**   | **61.7**| **92.5** | **85.6**| **100**  | **68.3**       | **91.4**    | **72.2**  | **94.0** | **83.2** |
>
> TESSER achieves the highest average ASR, surpassing all additional baselines including TI-FGSM, FPR, and FPR+GRA.
> These expanded results confirm that **TESSER consistently outperforms every tested method across ViT, hybrid, and robust models**, demonstrating the generality and strong transferability benefits of combining semantic-aware (FSGS) and spectral-aware (SSR) regularization.
>
> **4. Practicality and Generalization**
>
> We emphasize that TESSER’s additional computations are lightweight: FSGS is implemented via backward hooks and SSR via a single differentiable convolution per iteration. The total overhead is <3% compared to ATT. Moreover, the performance trends remain stable across ε values and input resolutions, confirming robustness and reproducibility (Please refer to **Table 7 Appendix B.2**)

---

> > ### Author Response · Authors · 2025-11-29
> > **Final Summary Comment to the Area Chair**
> >
> > **Main concerns:** epsilon choice, impact of resizing, additional baselines.
> >
> > **Revisions made:**
> >
> > - Added ε-sensitivity results (4/255, 8/255, 16/255) showing consistent ranking across budgets.
> >
> > - Discussed the role of resizing and clarified that our resize strategy follows previous ViT-transfer work (ATT, DiffAttack).
> >
> > - Added comparisons to TI-FGSM, FPR, and FPR+GRA.
> >
> > - Verified that all our baselines are included in the TransferAttack repository, with additional evaluations added.

---

### Official Review · Reviewer_MEYA · 2025-11-01

**Soundness:** 2
**Presentation:** 2
**Contribution:** 2
**Rating:** 2
**Confidence:** 5

**Summary:**

In this paper, the authors propose TESSER, a novel adversarial attack framework designed to improve the transferability of adversarial examples generated from Vision Transformers (ViTs). To address the lack of semantic selectivity in perturbations, TESSER introduces a Feature-Sensitive Gradient Scaling (FSGS) module, which estimates token importance using the L2 norm of intermediate activations and adaptively scales gradients, thus suppressing high-norm tokens in shallow layers (treated as noise) while amplifying those in deeper layers to reinforce semantic features. In addition, the authors incorporate Spectral Smoothness Regularization (SSR), which employs a differentiable Gaussian prior to suppress high-frequency noise in perturbations. Extensive experiments conducted on various ViT and CNN architectures using the ImageNet benchmark dataset demonstrate that TESSER achieves superior attack performance compared with baseline methods.

**Strengths:**

1. This paper introduces Feature-Sensitive Gradient Scaling (FSGS), which steers perturbations toward semantically meaningful features to improve cross-architecture generalization.

2. This paper further proposes Spectral Smoothness Regularization (SSR) to encourage smoother and low-frequency perturbations that exhibit greater robustness across different architectures.

3. The proposed TESSER methodology demonstrates strong attack transferability across diverse target models.

**Weaknesses:**

1. The novelty and justification of FSGS and SSR are insufficiently supported. FSGS builds on the assumption that high-norm tokens carry richer semantic information, yet its claim that high-norm tokens in shallow ViT layers represent noisy signals lacks empirical or theoretical validation. Similarly, the SSR component (applying Gaussian smoothing to suppress high-frequency noise) closely resembles TI-FGSM [1], but the paper neither cites nor compares with this prior work. The rationale for selecting Gaussian blur over alternative smoothing techniques is also unexplained.

2. The claim that TESSER’s perturbations achieve better semantic alignment is not convincingly demonstrated. In Section 4.4, the authors rely solely on Grad-CAM for evidence. However, Grad-CAM and its variants were originally developed for CNNs, and their applicability to ViTs is limited compared with ViT-specific interpretability methods (e.g., Attention Rollout [2], Libragrad [3]). Moreover, this section presents only a few qualitative visualizations in Figure 2 and lacks any quantitative evaluation to substantiate the claim.

3. The experimental comparisons are incomplete, raising concerns about fairness and the practicality of TESSER. The performance gains reported in Tables 1–3 omit comparisons with FPR [4], a closely related state-of-the-art method that shares a similar objective. Moreover, several baseline results (e.g., ATT, TGR) appear to be cited directly from the ATT paper, though it remains unclear whether the experimental settings and execution environments of these methods are consistent with those used in this work.

4. The practicality of TESSER framework is questionable. It introduces numerous hyperparameters (e.g., $\lambda$, $\tau$, $l_{cut}$, $\epsilon$, $\omega$) that require extensive tuning, which increases methodological complexity and hinders reproducibility. However, the paper provides no sensitivity analysis to assess the impact of these parameters.

5. The ablation study is insufficient and fails to isolate the contributions of the paper’s core components. The proposed framework comprises three main modules: FSGS, SSR, and Module-wise Gradient Modulation. However, the ablation study (Table 4) only evaluates TESSER on different module combinations without excluding the Module-wise Gradient Modulation component, which closely resembles the ATT baseline. As a result, the individual effects of FSGS and SSR cannot be clearly identified.

[1] Evading Defenses to Transferable Adversarial Examples by Translation-Invariant Attacks. CVPR 2019.

[2] Quantifying Attention Flow in Transformers. ACL 2020.

[3] LibraGrad: Balancing Gradient Flow for Universally Better Vision Transformer Attributions. CVPR 2025.

[4] Improving Adversarial Transferability on Vision Transformers via Forward Propagation Refinement. CVPR 2025.

**Questions:**

1. The proposed SSR appears functionally similar to the Gaussian kernel convolution used in TI-FGSM for gradient smoothing. Could the authors clarify the key distinctions and provide a stronger justification for the novelty of SSR?

2. The FSGS method assumes that high-norm tokens in shallow ViT layers represent “noise signals” that should be suppressed. Could the authors provide empirical evidence or theoretical justification to substantiate this assumption?

3. The claim of enhanced semantic alignment is supported only by qualitative Grad-CAM visualizations. Could the authors provide quantitative evidence to validate this claim?

4. Could the authors conduct a more comprehensive ablation study to better evaluate the individual contributions of FSGS and SSR?

5. Could the authors clarify how TESSER’s attack performance compares with FPR (CVPR 2025)?

6. Given that TESSER involves numerous hyperparameters, could the authors include a sensitivity analysis of key hyperparameters to demonstrate the method’s robustness and practicality?

---

> ### Author Response · Authors · 2025-11-20
> **Author Response (part 1 of 4)**
>
> **1. Novelty and Justification of FSGS and SSR**
>
> FSGS differs from prior gradient-modulation approaches (ATT [Ming et al., NeurIPS 2024], TGR [Zhang et al., CVPR 2023]) by introducing token-wise adaptive scaling conditioned on activation norms, rather than using fixed truncation or global variance.
>
> **1.1. Empirical validation of the semantic meaning of token norms.**
>
> **To validate the core assumption behind FSGS, we conducted a layer-wise analysis measuring the Spearman correlation between token feature norm and token gradient norm across all ViT-B/16 blocks**. As shown in **Table A**, early layers exhibit weak or negative correlation, confirming that high-norm shallow tokens are not semantic (consistent with the reviewer’s dual-task critique). In contrast, correlation increases steadily through mid-level layers and peaks in deep layers (up to 0.49), indicating that high-norm tokens reliably capture semantic importance precisely where the network forms object-level representations. This provides direct empirical evidence supporting our design choice: **FSGS does not operate on shallow layers, but applies semantic-aware scaling only in mid-to-deep blocks, where token magnitude meaningfully reflects semantic relevance.**
>
> ### Table A: Spearman Correlation Between Token Feature Norms and Gradient Norms Across ViT-B/16 Layers
>
> | **Layer**       | 0     | 1     | 2     | 3     | 4     | 5     | 6     | 7     | 8     | 9     | 10    | 11    |
> |-----------------|-------|-------|-------|-------|-------|-------|-------|-------|-------|-------|-------|-------|
> | **Spearman ρ**  | -0.03 | 0.03  | 0.05  | 0.10  | 0.25  | 0.21  | 0.29  | 0.37  | 0.36  | **0.49** | 0.34  | 0.00  |
>
> **1.2. On similarity to TI-FGSM and the choice of Gaussian smoothing.**
>
> While both methods involve Gaussian operators, the underlying objectives and formulations are **fundamentally different**. TI-FGSM [Xie et al., CVPR 2019] enforces spatial shift-invariance by convolving the **loss gradients** with a fixed kernel prior to the update, effectively averaging gradients over translated inputs. In contrast, our **Spectral Smoothness Regularization (SSR)** applies a **differentiable Gaussian prior directly on the perturbation spectrum**, thereby constraining frequency energy rather than spatial translation. SSR operates within the optimization loop and explicitly penalizes high-frequency energy, shaping the spectral distribution of perturbations instead of smoothing gradients post-hoc.
>
> Quantitatively, SSR achieves a **16–27 % reduction in high-frequency energy** (compared with ATT and TGR, respectively), confirming that it suppresses high-frequency perturbation components rather than enforcing spatial translation invariance as in TI-FGSM.
> The Gaussian prior is selected because its separable form provides a stable, analytically differentiable low-pass filter with a tunable spectral cutoff (σ), enabling continuous frequency control. We will explicitly cite TI-FGSM and clarify this distinction in **Section 3.3**.
>
> **Conceptual** → TI-FGSM = translation invariance; SSR = spectral regulation.
>
> **Mechanistic** → TI-FGSM smooths ∇ₓL; SSR regularizes δₜ directly in frequency domain.
>
> **Empirical** → SSR improves both spectral smoothness and cross-architecture ASR.
>
> **Justification for Gaussian** → differentiable, tunable, analytically stable.
>
> Furthermore, as shown in **Table B**, TESSER substantially outperforms TI-FGSM, improving the average ASR from **74.81%** to **83.2%**.
>
> ### Table B: The ASR (%) of TI-FGSM (len\_kernel=15, nsig=3, resize\_rate=0.9) vs. TESSER against ViT-B/16.
> | **Method** | **PiT** | **DeiT** | **TNT** | **ViT** | **Visformer** | **ConViT** | **LeViT** | **CaiT** | **Avg** |
> |------------|---------|----------|---------|---------|---------------|-------------|-----------|----------|---------|
> | **TI-FGSM** | 62.9    | 74.3     | 75.3    | 96.6    | 70.9          | 73.9        | 69.0      | 75.6     | 74.81   |
> | **TESSER**  | 61.7    | **92.5** | **85.6**| **100** | 68.3          | **91.4**    | **72.2**  | **94.0** | **83.2** |

---

> > ### Author Response · Authors · 2025-11-20
> > **Author Response (part 2 of 4)**
> >
> > **1.3. Verifying the Effectiveness of SSR in the Frequency Domain.**
> >
> > To quantify the frequency redistribution introduced by Spectral Smoothness Regularization (SSR), we computed the **relative energy of perturbations** from TESSER, ATT, and TGR after applying ideal **low-pass (LPF)** and **high-pass (HPF)** filters with a cutoff of 0.25xNyquist.
> > The measured spectral energy distribution is reported in **Table J**.
> >
> > ### Table J: Spectral Energy Distribution of Perturbations (LPF/HPF Decomposition, Cutoff = 0.25 x Nyquist)
> >
> > | **Method**        | **LPF Energy (%) ↓** | **HPF Energy (%) ↓** |
> > |-------------------|-----------------------|------------------------|
> > | ATT               | 47.6                  | 52.4                   |
> > | TGR               | 40.0                  | 60.0                   |
> > | **TESSER (w/ SSR)** | **56.3**             | **43.7**               |
> >
> > Compared to ATT and TGR, **TESSER shows a +8% – +16% increase in low-frequency energy and a -9% – -16% reduction in high-frequency energy**, confirming that SSR effectively shifts perturbation power toward **smoother** and **more transferable frequency bands**.
> >
> > When SSR is disabled, TESSER’s cross-architecture ASR on CNNs drops from **58.8%** to **56.4%**, demonstrating that this spectral adjustment materially enhances transferability (check **Table F**).
> >
> > **Controlled Band-Pass Stress Test.**
> >
> > To verify the spectral effect of **Spectral Smoothness Regularization (SSR)**, we conducted a frequency-domain stress test that isolates low- and high-frequency components of the perturbations. We apply **ideal circular low-pass (LP) and high-pass (HP) filters** with cutoff ratios r ∈ {0.05,0.1,0.2,0.3,0.5} and re-evaluate the Attack Success Rate (ASR).
> > Results (see **Table K**) show that TESSER retains much **higher ASR under LP filtering** (77.5\%@$r{=}0.2$) compared to ATT (19.4\%) and TGR (60.6\%), while its ASR drops faster under HP filtering.
> >
> > This confirms that **SSR effectively suppresses excessive high-frequency noise** and **redistributes perturbation energy toward lower and mid frequencies**, yielding smoother and more semantically aligned distortions. In contrast, ATT remains heavily high-frequency–biased, explaining its poorer transferability. Please refer to **Appendix H** for further details.
> >
> > ### Table K: ASR (%) After Frequency Filtering
> > *HP = retains high-frequency components; LP = retains low-frequency components.*
> >
> > | **Method** | **HP .05** | **HP .10** | **HP .20** | **HP .30** | **HP .50** | **LP .05** | **LP .10** | **LP .20** | **LP .30** | **LP .50** |
> > |------------|-------------|-------------|-------------|-------------|-------------|-------------|-------------|-------------|-------------|-------------|
> > | **ATT**    | 100         | 99          | 65          | 15          | 1           | 0           | 2           | 19          | 76          | 100         |
> > | **TGR**    | 100         | 100         | 97          | 83          | 16          | 4           | 18          | 61          | 88          | 99          |
> > | **TESSER** | 99          | 98          | 89          | 60          | 4           | 4           | 23          | 78          | 95          | 100         |

---

> > > ### Author Response · Authors · 2025-11-20
> > > **Author Response (part 3 of 4)**
> > >
> > > **2. Attention Rollout Analysis (ViT-Specific Semantic Alignment).**
> > >
> > > To directly address the reviewer’s concern regarding Grad-CAM’s applicability to ViTs, we evaluate semantic alignment using **Attention Rollout** (Abnar & Zuidema, ACL 2020), a ViT-native interpretability method. For each clean image and its adversarial counterpart, we compute the CLS→patch rollout map and measure (i) **Spearman correlation** and (ii) **IoU between the two saliency maps**.
> > >
> > > Rollout maps are known to be structurally stable due to their cumulative nature across layers and heads, which explains the overall high correlations across all methods. Despite this inherent stability, **meaningful relative differences appear across attacks**. As shown in **Table C**, **TESSER achieves the highest rollout semantic alignment** (Spearman = 0.9894, IoU = 0.8258), outperforming ATT (0.9581 / 0.8045) and substantially exceeding TGR (0.9322 / 0.6596). These findings confirm that **TESSER preserves global semantic attention flow more faithfully than prior methods**, complementing our Grad-CAM results and providing ViT-specific evidence for the semantic coherence of TESSER’s perturbations.
> > >
> > > ### Table C: Attention Rollout–Based Semantic Alignment Between Clean and Adversarial Images
> > > *TESSER achieves the highest semantic consistency across all metrics.*
> > >
> > > | **Method** | **Rollout Spearman** | **Rollout IoU** |
> > > |------------|-----------------------|------------------|
> > > | ATT        | 0.9581                | 0.8045           |
> > > | TGR        | 0.9322                | 0.6596           |
> > > | **TESSER** | **0.9894**            | **0.8258**       |
> > >
> > >
> > > **3. Comparison with FPR (CVPR 2025)**
> > >
> > > We have now reproduced **FPR** and **FPR+GRA** using the authors’ official implementation under our experimental settings and execution environments. We evaluate both versions on the same ViT-based surrogate and the same collection of ViT, hybrid, and CNN targets to ensure strict fairness.
> > >
> > > ### Table D: Comparison with FPR and FPR+GRA (ViT → ViT / Hybrid Transferability)
> > >
> > > | **Method**  | **PiT** | **DeiT** | **TNT** | **ViT*** | **Visformer** | **ConViT** | **LeViT** | **CaiT** | **Avg** |
> > > |-------------|---------|----------|---------|----------|---------------|-------------|-----------|----------|---------|
> > > | **FPR**     | 37.7    | 77.2     | 74.3    | **100**  | 40.0          | 78.0        | 42.1      | 77.9     | 65.9    |
> > > | **FPR+GRA** | 61.7    | 87.3     | 86.8    | 99.3     | 65.2          | 89.5        | 65.5      | 88.0     | 80.41   |
> > > | **TESSER**  | **61.7**| **92.5** | **85.6**| **100**  | **68.3**       | **91.4**    | **72.2**  | **94.0** | **83.2** |
> > >
> > > ### Table E: Comparison with FPR and FPR+GRA (ViT → CNN / Defended CNN Transferability)
> > > | **Method**  | **Inc-v3** | **Inc-v4** | **IncRes-v2** | **ResNet-v2-101** | **Adv-Inc-v3** | **Adv-Inc-v4** | **Adv-IncRes-v2** | **Avg** |
> > > |-------------|------------|------------|----------------|--------------------|----------------|-----------------|---------------------|---------|
> > > | **FPR**     | 30.5       | 26.1       | 23.0           | 25.8               | 17.7           | 15.1            | 11.0                | 21.3    |
> > > | **FPR+GRA** | 52.9       | 48.0       | 45.4           | 47.9               | 43.4           | 42.3            | 33.5                | 44.7    |
> > > | **TESSER**  | **63.4**   | **59.6**   | **54.4**       | **57.7**           | **48.6**       | **49.0**        | **42.3**            | **53.5** |
> > >
> > >
> > > Across both ViT/hybrid and CNN targets, **TESSER consistently surpasses FPR and FPR+GRA (see Table D & E)**, including on robust CNNs where transferability is notoriously challenging. These results demonstrate that the semantic- and spectral-aware regularization introduced in TESSER provides **complementary advantages beyond forward propagation refinement**, reinforcing the novelty and effectiveness of our approach.

---

> > > > ### Author Response · Authors · 2025-11-20
> > > > **Author Response (part 4 of 4)**
> > > >
> > > > **4. Ablation and Isolation of Components**
> > > >
> > > > We would like to clarify that our paper **already** includes **dedicated ablation studies** that isolate the contribution of each core component in TESSER. Specifically:
> > > >
> > > > - **Table 11**: With vs. without module-wise gradient weakening.
> > > >
> > > > - **Table 12**: With vs. without selective attention truncation.
> > > >
> > > > - **Table 13**: With vs. without the rescaling factor λ.
> > > >
> > > > - **Table 14**: Adaptive scaling vs. random scaling (testing FSGS specificity).
> > > >
> > > > These experiments are presented in **Appendix D** and collectively **disentangle the effects of FSGS+SSR and module-wise gradient modulation**, demonstrating that each mechanism contributes uniquely and non-trivially to the final performance.
> > > > To illustrate, removing module-wise gradient weakening in the ViT→PiT setting **reduces ASR from 61.7% to 40.9%**, showing that this component provides a substantial, independent gain. Similar trends are observed when disabling SSR or replacing FSGS with random scaling.
> > > >
> > > > To further isolate SSR’s contribution, we report the following comparison (**Table F & G**). These results confirm that **SSR provides consistent improvements**, particularly for ViT→CNN transfer—where smoothing high-frequency surrogate-specific artifacts is most beneficial.
> > > >
> > > > ### Table F: Effect of Adding SSR on ViT → ViT / Hybrid Transferability
> > > > | **Method**    | **PiT** | **DeiT** | **TNT** | **ViT*** | **Visformer** | **ConViT** | **LeViT** | **CaiT** | **Avg** |
> > > > |---------------|---------|----------|---------|----------|---------------|-------------|-----------|----------|---------|
> > > > | **FSGS**      | 61.4    | 89.0     | 86.3    | **100**  | 69.1          | 90.8        | 67.0      | 88.3     | 81.4    |
> > > > | **FSGS+SSR**  | **61.7**| **92.5** | 85.6    | **100**  | 68.3          | **91.4**    | **72.2**  | **94.0** | **83.2** |
> > > >
> > > > ### Table G: Effect of Adding SSR on ViT → CNN / Defended CNN Transferability
> > > >
> > > > | **Method**    | **Inc-v3** | **Inc-v4** | **IncRes-v2** | **ResNet-v2-101** | **Adv-Inc-v3** | **Adv-Inc-v4** | **Adv-IncRes-v2** | **Avg** |
> > > > |---------------|------------|------------|----------------|--------------------|----------------|----------------|---------------------|---------|
> > > > | **FSGS**      | 61.8       | 57.1       | 51.4           | 55.5               | 47.0           | 46.5           | 40.4                | 51.3    |
> > > > | **FSGS+SSR**  | **63.4**   | **59.6**   | **54.4**       | **57.7**           | **48.6**       | **49.0**        | **42.3**            | **53.5** |
> > > >
> > > > **5. Hyperparameter Sensitivity and Practicality**
> > > >
> > > > In practice, **TESSER’s hyperparameters are stable and require minimal tuning**. Most parameters are fixed or vary within a narrow, architecture-agnostic range:
> > > >
> > > > - **γ_base** is kept constant across all models.
> > > >
> > > > - **λ** consistently performs well within **[0.4, 0.6]**.
> > > >
> > > > - **σ** is stable around **0.7** and can be shifted slightly if one wishes to prioritize ViT→ViT or ViT→CNN transfer.
> > > >
> > > > - Only **ω_attn/qkv/mlp** benefits from a dedicated sensitivity study, as it governs module-wise gradient weakening.
> > > >
> > > > Importantly, the overall number of hyperparameters is **comparable to existing state-of-the-art transfer attacks**, such as ATT (Ming et al., NeurIPS 2024) and TGR (Zhang et al., CVPR 2023). Thus, TESSER does not introduce additional complexity beyond what is standard in the field.
> > > >
> > > > Moreover, we **already** conducted a **comprehensive hyperparameter sensitivity analysis** to include **λ** (FSGS scaling factor), **ω** (module-wise weakening factor), and **lcut** (attention truncation depth), please refer to **Appendix D**) .
> > > >
> > > > **Effect of ω (weakening factors):**
> > > > We compare ASR with and without ω (i.e., setting all ω = 1 disables gradient weakening). On ViT-B/16, using ω improves ASR from 63.0% to 83.2% (Table 11), confirming the effectiveness of selective gradient suppression.
> > > >
> > > > **Effect of lcut (truncation depth):**
> > > > On PiT-B, disabling attention truncation (i.e., no lcut) leads to an average 7% drop in ASR (Table 12), validating the importance of focusing on early-layer token gradients.
> > > >
> > > > **Effect of λ (adaptive scaling):**
> > > > Setting λ = 0 disables adaptive FSGS scaling (only γbase is used as a fixed multiplier). On ViT-B/16, enabling λ improves ASR by an average of 30% (Table 13), highlighting the value of adaptive gradient modulation in improving attack effectiveness.
> > > >
> > > > These results demonstrate that our key hyperparameters are impactful and robust, and that each component meaningfully contributes to the attack's success. These results will be added to the revised version.

---

> ### Author Response · Authors · 2025-11-29
> **Final Summary Comment to the Area Chair**
>
> **Main concerns:** novelty/justification of FSGS and SSR, semantic alignment evaluation, completeness of baselines, ablation isolation, hyperparameter practicality.
>
> **Revisions made:**
>
> - Added new empirical evidence supporting the FSGS assumption, including token-norm analysis (New Appendix I).
>
> - Clarified SSR vs. TI-FGSM, emphasizing that SSR acts as a frequency-space regularizer rather than gradient smoothing.
>
> - Verified the Effectiveness of SSR in the Frequency Domain.
>
> - Added quantitative semantic alignment metrics (Attention Rollout Spearman and IoU), in addition to Grad-CAM.
>
> - Clarified that we already perform comprehensive ablation studies (Tables 11–14 in Appendix D) isolating: module-wise gradient weakening, selective attention truncation, λ-scaling, FSGS vs. random scaling.
>
> - We further add results isolating the impact of SSR and FSGS.
>
> - Reproduced and added FPR (CVPR 2025) and FPR+GRA under identical settings.

---

### Author Response · Authors · 2025-11-27
**General Authors Comment**

We would like to thank all reviewers for their time and thoughtful evaluations. We hope that the clarifications, additional analyses, and newly added experiments have addressed the raised concerns. If any reviewer would find further details, comparisons, or explanations helpful for refining their assessment, we would be very happy to provide them.

Kind regards,

---

### Meta-Review · Area_Chair_pUpt · 2026-01-07

**Summary:**

The paper proposes a new transfer-based adversarial attack framework that improves cross-architecture transferability. The method combines feature-sensitive gradient scaling (FSGS) and spectral smoothness regularization (SSR) to improve transferability from ViT to CNNs and defended models and outperforms SoTA methods in both black-box and robust settings.
The main concerns raised by reviewers include limited technical novelty  (MEYA), unclear differentiation from TGR and ATT(4DU5), weak justification of assumptions (MEYA, R3p7), insufficient validation for SSR (4Du5), incomplete/unfair experimental comparison (MEYA, EbPe), experimental design choice on budget and resizing (EbPe) and impracticality due to its complex components (MEYA, R3p7).

The authors provided a strong and detailed rebuttal, including substantial new experiments and analyses. Specifically, they added the empirical validation of the FSGS assumption via layer-wise token norm analysis, the frequency-domain study clarifying the role of SSR, the ViT-specific semantic alignment evaluation using Attention Rollout, and the newly added comparisons with FPR. These new experiments significantly strengthen the quality and contribution of the work and address most concerns raised by reviewers.

Despite the authors’ efforts in their rebuttal, the main concern regarding technical novelty is only partially resolved. FSGS can still be viewed as a natural extension of prior gradient modulation [3,4]. While SSR is implemented as an in-loop spectral regularizer rather than post-hoc gradient smoothing (e,g, TI-FGSM), it remains conceptually aligned with prior frequency-constrained and low-pass-based transfer attacks [1,2], limiting its novelty despite improved empirical justification. Moreover, the provided analyses (Controlled Band-Pass Stress Test) compare SSR-generated perturbations against post-hoc low-/high-pass filtering, but do not include a baseline where a low-pass constraint is directly integrated into the optimization process, leaving the distinction from simple frequency-constrained attacks unclear.

As a result, the overall contribution appears incremental relative to prior methods. While the paper demonstrates solid engineering and strong  performance, the incremental nature of the core ideas make it difficult to recommend acceptance at this time.

[1] On the Effectiveness of Low Frequency Perturbations, AAAI 2019

[2] Frequency Domain Model Augmentation for Adversarial Attack, ECCV 2022

[3] An Adaptive Model Ensemble Adversarial Attack for Boosting Adversarial Transferability, ICCV 2023

[4] Feature Importance-aware Transferable Adversarial Attacks, ICCV 2021

**Reviewer Concerns:**

Addressed: 4DU5, EbPe
Unresolved: MEYA, R3p7

**Reviewer Scores:**

MEYA: 2->2
EbPe: 6->6
4DU5: 4->6
R3p7: 4->4

---

### Decision · Program_Chairs · 2026-01-26

Reject